# Parametric Prior Mapping Framework for Non-stationary Probabilistic Time Series Forecasting

Jinglin Li [* 1]   Jun Tan [* 1]   QI Fang [1]   Ning Gui [1]

## Abstract

Effectively modeling non-stationary dynamics in probabilistic multivariate time series(MTS) forecasting requires balancing expressiveness with robustness. Existing parametric approaches benefit from strong inductive biases but lack flexibility, whereas deep generative models struggle to capture complex temporal dependencies without extensive data and computation. We introduce Parametric Prior Mapping (PPM), a framework that injects parametric structural priors into a generative modeling process. Specifically, PPM utilizes a parametric estimator to derive a dynamic, adaptive prior that guides the learning of a complex predictive distribution via a learnable mapping. This design allows the model to retain the efficiency of parametric methods while exploiting the expressive power of generative models. Trained with a hybrid objective, PPM yields precise forecasts with well-calibrated uncertainty estimates. Empirical results show that PPM outperforms existing baselines in handling non-stationary data, offering a superior trade-off between accuracy and computational efficiency. The code is available at https://github.com/ljl8336/PPM.

## 1. Introduction

While conventional MTS forecasting focuses on point estimates, probabilistic MTS forecasting provides a comprehensive characterization of predictive uncertainty. This capability is pivotal for decision-making in high-stakes domains like weather (Price et al., 2025), finance (Yates et al., 1991), and transportation (Cheng et al., 2024) where the integrity and calibration of uncertainty estimates are as critical as point-wise accuracy. However, the task becomes significantly more complex in the presence of non-stationary dynamics, which necessitate the accurate tracking of evolving data distributions and time-varying stochasticity.

Existing methods generally fall into two categories: parametric approaches and deep generative models. Parametric methods define a specific likelihood for future trajectories and learn distribution parameters to quantify uncertainty, e.g., fixed Gaussian likelihood in DeepAR (Salinas et al., 2020) and a time-varying error covariance matrix for consecutive time series segments in BetterDeepAR (Zheng et al., 2024). While these models offer stability, scalability, and practical interpretability, they are generally constrained by the inconsistency between the assumed output family and the complex patterns under nonstationarity. By contrast, deep generative models learn the predictive distribution more flexibly, often without restrictive output assumptions. In recent years, many generative models have been proposed, e.g., denoising diffusion (Rasul et al., 2021) models and flow-based models (Kollovieh et al., 2025). These methods generally quantify uncertainty by learning to map simple noise into realistic future samples, allowing them to better capture the diverse range of possible future trajectories.

Theoretically, deep generative models can learn a nonparametric mapping to the prediction distribution from any prior distribution (Lu & Lu, 2020). However, in non-stationary time series with finite samples and limited computational power, the form of the prior can substantially affect the reachability of the sampling trajectory and its modeling capabilities (Lee et al., 2024). To address this problem, recent models introduce designs to better approximate the true prediction distribution. TMDM (Li et al., 2024) uses $\mathcal{N}(f(\boldsymbol{x}), \mathbf{I})$ as an endpoint, with $f(\boldsymbol{x})$ representing changing averages with limitations of its fixed unit deviation $\mathbf{I}$. To address this issue, NsDiff (Ye et al., 2025) employs a fixed-length sliding window to compute changing variance as a deviation prior.

Although existing diffusion-based methods like TMDM and NsDiff attempt to address prior mismatch, they fundamentally lack the flexibility to handle dynamic shifts in data distribution. Figure 1 highlights this limitation using the Traffic dataset. We observe a strong correlation between

*Equal contribution  [1]School of Computer Science and Engineering, Central South University, Changsha, China. Correspondence to: QI Fang <csqifang@csu.edu.cn>, Ning Gui <ninggui@gmail.com>.

*Proceedings of the 43$^{rd}$ International Conference on Machine Learning*, Seoul, South Korea. PMLR 306, 2026. Copyright 2026 by the author(s).

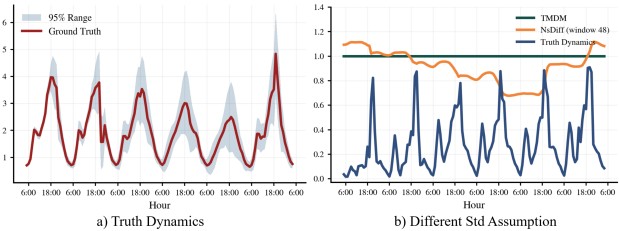

*Figure 1.* Comparisons of source distribution from different baselines for the Traffic dataset. The top figure shows the true value and deviation, as during rush hour, truth dynamics have more uncertainty, and during the middle night, the traffic is rather steady. However, the deviation previously used/calculated by TMDM and NsDiff is far from this fact.

traffic volume and variance: stable, low-volume periods at the daily trough (typically 5∼6AM) contrast sharply with volatile, congested rush hours(5∼6PM). The right panel demonstrates that both TMDM and NsDiff struggle to adapt to these fluctuations. By enforcing a rigid initialization scheme, these methods fail to model the time-varying aleatoric uncertainty, resulting in miscalibrated predictive distributions.

Building upon these insights, it is evident that parametric and generative approaches possess complementary strengths. While parametric models provide an efficient inductive bias for uncertainty estimation, generative models offer the necessary expressivity to capture complex, non-stationary dynamics. To this end, we propose Parametric Prior Mapping (**PPM**), a novel framework that synergistically integrates these two paradigms. Specifically, PPM leverages the fast estimation capabilities of parametric methods to characterize input distributions, from which an adaptive conditional prior is derived. We then learn a mapping that transforms this structured prior into a flexible sample-based predictive distribution. To ensure both point-wise precision and distributional robustness, we optimize PPM using a hybrid objective that combines Negative Log-Likelihood (NLL) with Mean Squared Error (MSE), effectively balancing predictive calibration with trajectory accuracy. Our contributions are summarized as follows:

- **Conceptual Synergy**: We propose PPM, a novel end-to-end framework that synergizes parametric estimation with deep generative modeling. By bridging the gap between rigid parametric priors and flexible generative outputs, PPM captures complex, non-stationary dynamics without relying on manually-tuned heuristic priors.

- **Technical Innovation**: We introduce a parametric push-forward mechanism to rapidly derive an adaptive conditional prior from input sequences, coupled with a mapping module that refines resampled prior distributions into target trajectories. To ensure training stability and distri-

butional fidelity, a hybrid objective integrating NLL and MSE is proposed, enabling the model to track evolving uncertainties while maintaining point-wise precision.

- **Empirical Excellence**: Extensive evaluations across diverse benchmarks demonstrate that PPM significantly outperforms six state-of-the-art baselines, achieving up to a 31.2% reduction in CRPS and 44.3% in QICE compared to the second-best model. Notably, PPM delivers a $2\times$ to $100\times$ speedup in inference relative to leading diffusion-based models, highlighting its superior trade-off between predictive accuracy and computational efficiency.

## 2. Related Work

### 2.1. Parametric Probabilistic Forecasting

Parametric probabilistic forecasting typically assumes explicit likelihood families (e.g., Gaussian) and minimizes negative log-likelihood (NLL). DeepAR (Salinas et al., 2020) employs autoregressive RNNs to predict per-step parameters. To improve correlation modeling, (Zheng et al., 2024) relaxes i.i.d. assumptions to learn time-varying error structures, while Zheng & Sun (2024) captures cross-series dependencies via structured covariance. Hybrid approaches like DeepState (Rangapuram et al., 2018) and Deep Factors (Wang et al., 2019) integrate deep networks with structured models to enhance interpretability. In parallel, quantile regression offers a distribution-free alternative: MQ-RNN (Wen et al., 2017) generates multi-horizon quantiles via seq2seq architectures, and TFT (Lim et al., 2021) utilizes pinball losses for calibrated intervals, with recent advances extending this to unified *any-quantile* forecasting (Smyl et al., 2024).

### 2.2. Deep Generative Model

In recent years, Deep Generative Models have received increasing attention and witnessed significant advancements in Probabilistic Time Series Forecasting. TimeGrad (Rasul et al., 2021) combines the autoregressive method and a diffusion probabilistic model to better model high-dimensional distributions. D3VAE (Li et al., 2022) is another autoregressive model that incorporates a bidirectional Variational Autoencoder and component separation to achieve Temporal Data Augmentation. TimeDiff (Shen & Kwok, 2023) leverages a diffusion process on the entire sequence to generate outputs based on given conditions. Furthermore, many models effectively exploit conditional information to facilitate the learning of generative models. TMDM (Li et al., 2024) integrates conditional information into the diffusion forward process with a Transformer model to enhance the estimation of uncertainty distributions over sequences. NsDiff (Ye et al., 2025) additionally incorporates an explicit variance estimation module to parameterize the prior uncertainty,

trained under supervision from variances computed over sliding-window segments. D3U (Li et al., 2025b) simplifies the forecasting problem by modeling residuals to avoid non-stationarity; however, this makes its performance heavily dependent on the quality of the required point predictor.

# 3. Background

## 3.1. Probabilistic Time Series Forecasting

Given a historical multivariate time series $\boldsymbol{x} \in \mathbb{R}^{H \times C} = \{x_1, x_2, ..., x_C | x_c \in \mathbb{R}^H\}$, probabilistic MTS forecasting aims to estimate the conditional distribution of the future trajectory $p(\boldsymbol{y}|\boldsymbol{x})$, where $\boldsymbol{y} \in \mathbb{R}^{L \times C} = \{y_1, y_2, ..., y_C | y_c \in \mathbb{R}^L\}$. Let $H$, $L$, and $C$ denote the lookback window, prediction horizon, and variable count, respectively.

Directly modeling the exact density of this high-dimensional distribution is often intractable. Consequently, state-of-the-art sampling-based methods approximate the target distribution $p_\theta(\boldsymbol{y}|\boldsymbol{x})$ using a model parameterized by $\theta$. The final probabilistic forecasts take the form of an empirical Monte Carlo approximation, represented by a set of $K$ generated samples $\hat{\mathcal{Y}} = \{\hat{\boldsymbol{y}}^{(i)}\}_{i=1}^K$, where each $\hat{\boldsymbol{y}}^{(i)} \sim p_\theta(\cdot|\boldsymbol{x})$.

## 3.2. Push-forward Distribution

We formulate the predictive distribution using the *push-forward* measure, which formalizes how a probability distribution transforms under a deterministic mapping.

Let $\boldsymbol{z}$ be a latent variable with prior $p_{\boldsymbol{z}}$ and $T : \mathcal{Z} \rightarrow \mathcal{Y}$ be a measurable map. The *push-forward distribution* $p_{\boldsymbol{y}} = T_\# p_{\boldsymbol{z}}$ is the distribution of $\boldsymbol{y} = T(\boldsymbol{z})$, formally satisfying:

$$(T_\# p_{\boldsymbol{z}})(A) := p_{\boldsymbol{z}}\left(T^{-1}(A)\right), \quad \forall A \subseteq \mathcal{Y}. \tag{1}$$

This framework unifies modern generative models like Normalizing Flows (Papamakarios et al., 2021) and Diffusion (Ho et al., 2020), which essentially learn a transport map from a prior to the data distribution. While theoretically possessing universal approximation capabilities (Villani et al., 2008), learning such maps from generic noise under limited compute is suboptimal. Motivated by information-theoretic insights (Xu et al., 2020), we propose using a structure-aware prior rather than generic noise to ease the transport complexity and improve expressivity.

# 4. PPM Method

In this section, we present the overall end-to-end achitecture of PPM. The core intuition behind PPM is to construct a high-fidelity conditional predictive distribution by refining a tractable parametric prior via a learnable push-forward mapping. The workflow is illustrated in Figure 2.

## 4.1. Training Scheme and Inference

**Training Stage** The training pipeline of PPM decomposes into three sequential phases: 1) **Parametric Prior Induction:** Given the historical context, we first employ an encoder to infer the sufficient statistics (e.g., mean and variance) of a parametric distribution. We then resample from this distribution to generate an initial set of latent samples, effectively constructing a context-aware prior (corresponding to the *upper-left* module in Figure 2). 2) **Distributional Push-forward:** To capture complex, non-Gaussian dynamics that simple parametric forms cannot model, we push the resampled prior through a learnable non-linear mapping. This step transforms the coarse prior into an expressive, sample-based output distribution, as depicted in the *upper-right* panel of Figure 2. 3) **Hybrid Objective Optimization:** To align the generated samples with the ground truth, we employ a hybrid loss function. Specifically, we utilize Kernel Density Estimation (KDE) to approximate the continuous probability density of the generated samples, allowing for the minimization of the Negative Log-Likelihood (NLL). This is augmented by an averaged MSE term to ensure trajectory consistency. This optimization process (illustrated in the *lower-right* panel) encourages the output distribution to accurately approximate the true predictive density.

**Inference Stage.** Once trained, PPM serves as an efficient generative forecaster. During inference, the model directly outputs a sample-based predictive distribution by pushing the conditional prior through the learned mapping. Notably, the size of the generated ensemble is determined solely by the number of draws, allowing for flexible trade-offs between estimation precision and computational cost.

## 4.2. Parametric Prior Induction

To initiate the generative process, we first construct a tractable conditional prior distribution $p_\theta(\boldsymbol{z}|\boldsymbol{x})$ that encapsulates the stochastic properties of the historical context.

**Backbone Encoder.** We employ a temporal encoder $f_\theta(\cdot)$ to extract representations from the historical sequence $\boldsymbol{x}$. While our framework is *backbone-agnostic*: allowing for the integration of various architectures such as Transformers or RNNs. we instantiate $f_\theta(\cdot)$ as a lightweight Multi-Layer Perceptron (MLP) in this work to balance predictive capability with computational efficiency.

**Prior Parameterization.** Given the historical series $\boldsymbol{x}$, the encoder maps the input to the sufficient statistics of the prior distribution:

$$[\boldsymbol{\mu}, \boldsymbol{\sigma}] = f_\theta(\boldsymbol{x}), \quad \text{where } \boldsymbol{\mu}, \boldsymbol{\sigma} \in \mathbb{R}^{C \times D}. \tag{2}$$

Here, $D$ denotes the dimensionality of the latent space. Based on these parameters, we define the conditional prior

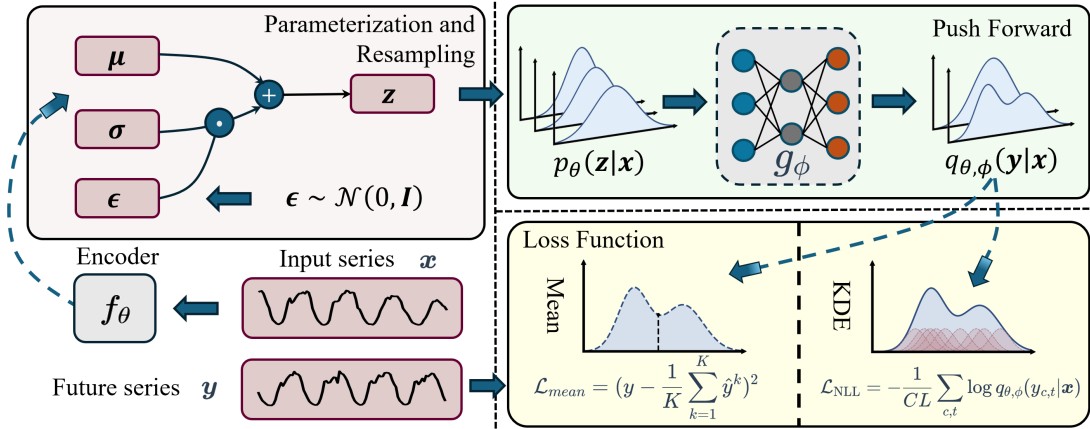

*Figure 2.* PPM is trained in three stages: encode historical context to estimate the prior's parameters and resample a sample-based prior; push the prior forward to obtain the predictive output distribution; then use KDE to estimate the conditional predictive density, minimizing averaged NLL (MLE) over the horizon with an auxiliary averaged MSE term.

as a factorized (diagonal) multivariate Gaussian:

$$p_\theta(\boldsymbol{z}|\boldsymbol{x}) = \mathcal{N}\left(\boldsymbol{z}; \boldsymbol{\mu}, \text{diag}(\boldsymbol{\sigma}^2)\right). \tag{3}$$

This parametric form allows for efficient sampling and gradient propagation (via the reparameterization trick).

### 4.3. Distributional Push-forward

To bridge the gap between the latent parametric space and the target data space, we employ the *reparameterization trick* (Kingma & Welling, 2013). This technique allows us to generate differentiable samples from the conditional prior while preserving the gradients for backpropagation. Specifically, we draw a set of $K$ latent samples $\{\boldsymbol{z}^{(k)}\}_{k=1}^K$ via the transformation:

$$\boldsymbol{z}^{(k)} = \boldsymbol{\mu} + \boldsymbol{\sigma} \odot \boldsymbol{\epsilon}^{(k)}, \quad \text{where } \boldsymbol{\epsilon}^{(k)} \sim \mathcal{N}(\boldsymbol{0}, \mathbf{I}). \tag{4}$$

Next, we define the predictive distribution as the push-forward of this sampled prior through a learnable mapping function $g_\phi : \mathbb{R}^{C \times D} \to \mathbb{R}^{L \times C}$. The induced predictive density is formally given by:

$$q_\phi(\boldsymbol{y}|\boldsymbol{x}) = (g_\phi)_\# p_\theta(\boldsymbol{z}|\boldsymbol{x}). \tag{5}$$

We obtain empirical samples of the forecast $\hat{\boldsymbol{y}}$ by applying the mapping to the latent draws: $\hat{\boldsymbol{y}}^{(k)} = g_\phi(\boldsymbol{z}^{(k)})$. The set of transformed samples $\{\hat{\boldsymbol{y}}^{(k)}\}_{k=1}^K$ constitutes the empirical approximation of the predictive distribution $q_{\theta,\phi}(\cdot|\boldsymbol{x})$.

**Architecture Design.** A key advantage of our framework is that the learned prior $p_\theta(\boldsymbol{z}|\boldsymbol{x})$ can encode rich, context-dependent semantic information (due to the over-complete latent dimension and end-to-end design). Consequently, the burden on the transport map is significantly reduced. We find that a lightweight channel-independent Multi-Layer Perceptron (MLP) is sufficient to realize this mapping. Specifically, we implement $g_\phi(\cdot)$ as a two-layer MLP with GeLU

activation, which projects the $D$-dimensional latent codes back to the $L$-dimensional forecast horizon.

### 4.4. Optimization Objective

PPM is optimized using a hybrid objective that balances probabilistic calibration with trajectory accuracy. The primary objective is to maximize the likelihood of the ground truth $\boldsymbol{y}$ under the learned predictive distribution.

**KDE-based NLL.** The Negative Log-Likelihood (NLL) for a single training instance $(\boldsymbol{x}, \boldsymbol{y})^{(i)}$ is defined as:

$$\mathcal{L}_{\text{NLL}}^{(i)} \triangleq -\frac{1}{CL} \sum_{c=1}^{C} \sum_{t=1}^{L} \log q_{\theta,\phi}(y_{c,t}|\boldsymbol{x}) \tag{6}$$

It's notable that we model the marginal distribution at each time step $p(y_{c,t})$, which does not imply that we assume different time steps to be independent or identically distributed. Since our model is implicit (outputting samples rather than density parameters), the exact likelihood is intractable. To enable gradient-based optimization, we employ Kernel Density Estimation (KDE) (Dehnad, 1987) to approximate the continuous probability density $q_{\theta,\phi}(y_{c,t}|\boldsymbol{x})$ from the discrete sample set $\{\hat{\boldsymbol{y}}^{(k)}\}_{k=1}^K$:

$$q_{\theta,\phi}(y_{c,t}|\boldsymbol{x}) \approx \hat{q}_h(y_{c,t}|\boldsymbol{x}) = \frac{1}{Kh} \sum_{k=1}^{K} \mathcal{K}\left(\frac{y_{c,t} - \hat{y}_{c,t}^{(k)}}{h}\right) \tag{7}$$

where $\hat{q}_h(y_{c,t}|\boldsymbol{x})$ is the density estimated by KDE, $h$ is the bandwidth parameter and $\mathcal{K}(\cdot)$ is the kernel function. Using a Gaussian kernel, this simplifies to:

$$q_{\theta,\phi}(y_{c,t}|\boldsymbol{x}) \approx \frac{1}{\sqrt{2\pi}Kh} \sum_{k=1}^{K} \exp\left(-\frac{(y_{c,t} - \hat{y}_{c,t}^{(k)})^2}{2h^2}\right). \tag{8}$$

In practice, we compute this efficiently in the log domain using the Log-Sum-Exp trick to ensure numerical stability.

**Auxiliary Mean MSE Regularization.** Relying solely on KDE-based NLL can lead to optimization instability, particularly in the early stages of training. If the generated samples are far from the ground truth, the Gaussian kernel values become negligible, causing gradients to vanish. To mitigate this and provide a stable, global gradient signal, we incorporate a Mean Squared Error (MSE) term based on the sample's mean:

$$\mathcal{L}_{\text{MM}}^{(i)} = \left\| \boldsymbol{y} - \frac{1}{K} \sum_{k=1}^{K} \hat{\boldsymbol{y}}^{(k)} \right\|^2. \tag{9}$$

This term acts as a *mean-seeking* anchor, forcing the predictive distribution to center around the ground truth trajectory.

**Final Loss.** The total objective function is a weighted sum of the probabilistic and deterministic losses:

$$\mathcal{L}_{\text{total}} = \alpha \cdot \mathcal{L}_{\text{NLL}} + \mathcal{L}_{\text{MM}}, \tag{10}$$

where $\alpha$ is a hyperparameter balancing distribution sharpness and point-wise accuracy.

## 5. Theoretical Properties of PPM

In this section, we further analyze our method from a theoretical perspective, focusing on consistency and finite-sample errors, the expressiveness of the push-forward conditional distribution, and the gradient/optimization properties that underpin training stability. Full details of the theory can be found in Appendix A.

**Theoretical Setup:** For clarity we analyze a single scalar coordinate $(c, t)$ and omit subscripts when no confusion arises. Given history $\boldsymbol{x}$, the model induces a conditional distribution over forecasts via a push-forward map $\hat{y} = g_\phi(z)$ with $z \sim p_\theta(\cdot | \boldsymbol{x})$ (e.g., reparameterized Gaussian). Let $q_{\theta,\phi}(y | \boldsymbol{x})$ denote the corresponding conditional density.

Given $K$ i.i.d. samples $\{\hat{y}^k\}_{k=1}^{K} \sim q_{\theta,\phi}(\cdot | \boldsymbol{x})$, PPM uses KDE:

$$\hat{q}_h(y | \boldsymbol{x}) = \frac{1}{Kh} \sum_{k=1}^{K} \mathcal{K}\left( \frac{y - \hat{y}^k}{h} \right), \tag{11}$$

and computes the log-likelihood in a numerically stable manner via log-sum-exp (Gaussian kernel), followed by a floor:

$$\log \tilde{q}_h(y | \boldsymbol{x}) := \max\{\log \hat{q}_h(y | \boldsymbol{x}), \log \varepsilon\}, \quad \varepsilon > 0. \tag{12}$$

This truncation prevents extremely small estimated densities from causing unstable gradients.

**Consistency and finite-$(K, h)$ error decomposition:** Let $q_h(\cdot | \boldsymbol{x}) := (q_{\theta,\phi}(\cdot | \boldsymbol{x}) * \mathcal{K}_h)(\cdot)$ be the smoothed density. Under standard KDE assumptions (bounded kernel with finite second moment and twice differentiable $q_{\theta,\phi}$), the truncated NLL admits an explicit decomposition into a finite-$K$ stochastic term and a smoothing-bias term.

**Theorem 5.1** (Finite-$(K, h)$ NLL error bound (informal)). *Let $\hat{\mathcal{L}}_{\text{NLL}}^{(i)} = -\frac{1}{CL} \sum_{c,t} \log \max\{\hat{q}_h(y_{c,t} | \boldsymbol{x}), \varepsilon\}$. Then for any $\delta \in (0, 1)$, with probability at least $1 - \delta$,*

$$\left| \hat{\mathcal{L}}_{\text{NLL}}^{(i)} - \mathcal{L}_{\varepsilon}^{(i)} \right| \lesssim \underbrace{\frac{h^2}{\varepsilon}}_{\text{smoothing bias}} + \underbrace{\frac{\sqrt{\log(CL/\delta)}}{\varepsilon h \sqrt{K}}}_{\text{finite-}K \text{ fluctuation}}, \tag{13}$$

*where*

$$\mathcal{L}_{\varepsilon}^{(i)} := -\frac{1}{CL} \sum_{c,t} \log \max\{q_{\theta,\phi}(y_{c,t} | \boldsymbol{x}), \varepsilon\}. \tag{14}$$

*Proof.* Full proof in Appendix A.1. This bound clarifies the role of KDE in PPM: the training objective is not only consistent in the ideal limit, but also admits an explicit, controlled approximation error at finite $(K, h)$. In particular, Eq. (13) implies consistency: letting $h \to 0$ and $Kh^2 \to \infty$ drives both the $O(h^2/\varepsilon)$ smoothing bias and the $O(1/(\varepsilon h \sqrt{K}))$ finite-$K$ fluctuation to zero, hence $\hat{\mathcal{L}}_{\text{NLL}}^{(i)} \xrightarrow{p} \mathcal{L}_{\varepsilon}^{(i)}$. Moreover, Eq. (13) reveals a practical trade-off: smaller $h$ sharpens the likelihood but amplifies finite-$K$ noise via $1/(h\sqrt{K})$, while larger $h$ reduces variance at the expense of an $O(h^2)$ smoothing bias. The bound shows that $K$ and $h$ must be co-tuned to ensure stable training.

**Expressiveness of push-forward:** For each history $\boldsymbol{x}$, PPM generates forecasts via a push-forward map $\hat{y} = g_\phi(z)$ with $z \sim p_\theta(\cdot | \boldsymbol{x})$. Under mild regularity of the target conditional distribution and the universal approximation property of MLPs, this conditional push-forward family is dense in $W_1$ (and thus in the weak topology with first-moment control).

**Theorem 5.2** (Universality of conditional push-forward (informal)). *Assume the ground-truth conditional distribution $p^*(\cdot | \boldsymbol{x})$ has a finite first moment. Then for any $\epsilon > 0$ and fixed $\boldsymbol{x}$, there exists parameters $\phi$ such that the generated conditional law*

$$q_\phi(\cdot | \boldsymbol{x}) := (g_\phi)_\# p_\theta(\cdot | \boldsymbol{x})$$

*satisfies*

$$W_1\big(q_\phi(\cdot | \boldsymbol{x}), p^*(\cdot | \boldsymbol{x})\big) < \epsilon.$$

*Proof.* Full proof is deferred to Appendix A.2. Theorem 5.2 demonstrates the expressiveness of the push-forward parameterization: by learning the mapping $g_\phi$, the induced conditional distribution $(g_\phi)_\# p_\theta(\cdot | \boldsymbol{x})$ can approximate highly complex prediction distributions in $W_1$.

**Gradient Structure and Stabilization:** For the Gaussian kernel, define $s_k(y) = \exp\left(-\frac{(y-\hat{y}^{(k)})^2}{2h^2}\right)$ and responsibilities $\bar{\omega}_j(y) = \frac{s_j(y)}{\sum_{k=1}^{K} s_k(y)}$. Ignoring the truncation for clarity, the per-coordinate KDE-NLL gradient w.r.t. sample $\hat{y}^j$ is

$$\frac{\partial}{\partial \hat{y}^{(j)}} \left[ -\log \hat{q}_h(y|\boldsymbol{x}) \right] = \frac{1}{h^2} \bar{\omega}_j(y) \left( \hat{y}^{(j)} - y \right). \quad (15)$$

With the log-truncation in (12), the gradient is further multiplied by $\mathbf{1}\{\hat{q}_h(y|\boldsymbol{x}) \geq \varepsilon\}$ (see Appendix A.3).

As $h$ decreases, $\bar{\omega}_j(y)$ concentrates on the nearest samples, yielding winner-take-all-like gradients and higher variance when $K$ is finite. PPM therefore adds an MM anchor on the sample mean $\bar{y} = \frac{1}{K}\sum_k \hat{y}^{(k)}$, whose gradient is dense:

$$\frac{\partial}{\partial \hat{y}^{(j)}} (y - \bar{y})^2 = \frac{2}{K}(\bar{y} - y). \quad (16)$$

Combining (15)–(16) yields complementary updates: The KDE-NLL term shapes distributional structure through responsibility-weighted residuals, while the MM term stabilizes training by consistently anchoring the first moment, especially in early stages with finite $(K, h)$. Full statements and proofs are provided in Appendix A.3.

# 6. Experiment

## 6.1. Experiment Setup

**Datasets:** Seven real-world datasets with diverse spatiotemporal dynamics were chosen, concluding ETTh1, ETTh2, ETTm1, ETTm2, Electricity, Traffic, and Weather. Table 1 presents basic statistical information about these datasets. We also report the average variance over the Fourier spectrum to examine the evolving dynamics (Ye et al., 2024) in the table, and larger variance implies more complex data dynamics. Further details can be found in Appendix B.1.

*Table 1.* Summary of dataset statistics.

| Dataset | Variables | Freq. | Length | Variance |
|---|---|---|---|---|
| ETTh1 | 7 | 1 Hour | 14307 | 3.690 |
| ETTh2 | 7 | 1 Hour | 14307 | 1.013 |
| ETTm1 | 7 | 15 Minutes | 69680 | 3.330 |
| ETTm2 | 7 | 15 Minutes | 69680 | 1.648 |
| Electricity | 321 | 1 Hour | 26304 | 2.645 |
| Traffic | 862 | 1 Hour | 17544 | 14.225 |
| Weather | 21 | 10 Minutes | 52696 | 0.387 |

**Baselines:** We select several strong forecasting baselines for comparison, including: DeepAR (Salinas et al., 2020), TimeGrad (Rasul et al., 2021), D3VAE (Li et al., 2022), TimeDiff (Shen & Kwok, 2023), DiffusionTs (Yuan & Qiao, 2024), TMDM (Li et al., 2024), and NsDiff (Ye et al., 2025). The description of these baselines can be found in Appendix B.2.

**Metrics:** The performance of probabilistic forecasting is evaluated with the Continuous Ranked Probability Score (CRPS) (Matheson & Winkler, 1976) and Quantile Interval Coverage Error (QICE) (Han et al., 2022). Additionally, point forecasting performance is assessed using MSE and Mean Absolute Error (MAE). For these metrics, smaller values indicate better performance. Detailed descriptions of these metrics can be found in Appendix B.3.

**Implementation details:** In the experiments, we set the lookback window size H 96 and prediction length L 192, respectively. The parameters of KDE-NLL loss, including bandwidth $h$ and loss weighting $\alpha$, are set to 0.3 and 0.1, respectively, for all datasets. The number of samples in triaining stage is set as 100. At inference, 100 samples are used to approximate the estimated distribution. For the baseline models, we utilize their default parameters. All experiments are implemented using PyTorch and executed on RTX A100. Further implementation details and sensitivity analysis are provided in Appendix B.4 and Appendix C.6.

## 6.2. Main Results

To evaluate the performance of PPM in probabilistic MTS forecasting, we conducted experiments on seven real-world datasets and compared it against several competitive baseline methods, and all experiments were repeated 5 times for the mean. The backbone for $f_\theta(\cdot)$ is an MLP combined. As summarized in Table 2 and Table 3, PPM consistently achieves state-of-the-art (SOTA) performance, demonstrating superior capabilities in distribution estimation and point forecasting. Notably, the improvement is more pronounced on traffic datasets with higher variability as shown in the Table 1, demonstrating PPM's capability to model evolving time series. We further evaluate alternative backbones and observe consistent gains with more expressive architectures; detailed results are provided in Appendix C.1.

## 6.3. Sample Showcases

Figure 3 visualizes PPM forecasts against the SOTA diffusion model NsDiff on two datasets. We observe that while NsDiff's sliding-window prior suffices for ETTm1 due to its locally stable statistics, it fails to adapt to the highly dynamic Traffic. In contrast, PPM accurately captures time-varying uncertainty patterns, such as increased variance during rush hours and consistently lower uncertainty at early morning in Traffic, validating its capability to model complex distributions. More cases can be found in Appendix D.

## 6.4. Analysis for PPM Framework

We dissect PPM's internal mechanisms following its generative pipeline. First, we validate the push-forward robustness in recovering complex distributions from Parametric priors.

*Table 2.* Performance comparisons on seven real-world datasets regarding CRPS and QICE(%). The **best**/second results are highlighted in **bold**/underline, respectively. Lower CRPS and QICE(%) values indicate better performance.

| Dataset | ETTh1 | | ETTh2 | | ETTm1 | | ETTm2 | | Weather | | Electricity | | Traffic | |
|---|---|---|---|---|---|---|---|---|---|---|---|---|---|---|
| Method | CRPS | QICE | CRPS | QICE | CRPS | QICE | CRPS | QICE | CRPS | QICE | CRPS | QICE | CRPS | QICE |
| DeepAR | 0.466 | 2.215 | 0.738 | 5.067 | 0.497 | 2.559 | 0.378 | 4.422 | 0.293 | 7.186 | 0.376 | 3.775 | 0.496 | 5.488 |
| TimeGrad | 0.636 | 5.031 | 0.990 | 7.921 | 1.001 | 8.097 | 0.644 | 2.744 | 0.362 | 2.954 | 0.405 | 5.788 | 0.694 | 6.546 |
| TimeDiff | 0.457 | 14.821 | 0.508 | 7.663 | 0.466 | 14.772 | 0.313 | 7.735 | 0.304 | 7.819 | 0.744 | 9.536 | 0.774 | 9.433 |
| D3VAE | 0.548 | 5.124 | 1.098 | 6.687 | 0.519 | 6.034 | 0.818 | 5.582 | 0.427 | 7.985 | 0.449 | 6.705 | 0.504 | 6.961 |
| DiffusionTS | 0.539 | 3.041 | 1.077 | 5.690 | 0.640 | 6.997 | 0.835 | 5.987 | 0.375 | **2.351** | 0.493 | 5.021 | 0.477 | 4.843 |
| TMDM | 0.468 | 2.354 | 0.382 | **1.955** | 0.364 | 2.024 | 0.322 | 1.843 | 0.240 | 3.674 | 0.462 | 10.995 | 0.562 | 10.873 |
| NsDiff | 0.417 | 1.564 | 0.349 | 2.125 | 0.336 | **1.174** | 0.270 | 2.343 | 0.253 | 5.134 | 0.286 | 7.595 | 0.367 | 8.366 |
| PPM | **0.337** | **1.481** | **0.306** | 2.011 | **0.314** | 1.782 | **0.237** | **1.580** | **0.215** | 2.684 | **0.206** | **2.435** | **0.252** | **2.744** |

*Table 3.* Performance comparison on seven real-world datasets based on MSE and MAE. The **best**/second results are highlighted in **bold**/underline, respectively. Lower MSE and MAE values indicate better performance.

| Dataset | ETTh1 | | ETTh2 | | ETTm1 | | ETTm2 | | Weather | | Electricity | | Traffic | |
|---|---|---|---|---|---|---|---|---|---|---|---|---|---|---|
| Method | MSE | MAE | MSE | MAE | MSE | MAE | MSE | MAE | MSE | MAE | MSE | MAE | MSE | MAE |
| DeepAR | 0.797 | 0.624 | 1.800 | 0.992 | 0.870 | 0.697 | 0.535 | 0.508 | 0.259 | 0.333 | 0.304 | 0.377 | 0.968 | 0.524 |
| TimeGrad | 1.232 | 0.909 | 2.449 | 1.230 | 2.779 | 1.190 | 1.101 | 0.811 | 0.562 | 0.510 | 0.460 | 0.477 | 1.647 | 0.814 |
| TimeDiff | 0.493 | 0.473 | 0.544 | 0.527 | 0.539 | 0.480 | 0.286 | 0.352 | 0.266 | 0.316 | 0.836 | 0.758 | 1.422 | 0.788 |
| D3VAE | 0.906 | 0.707 | 3.392 | 1.520 | 0.815 | 0.637 | 2.106 | 1.140 | 0.445 | 0.449 | 0.520 | 0.528 | 1.097 | 0.577 |
| DiffusionTS | 0.917 | 0.714 | 2.941 | 1.318 | 1.036 | 0.755 | 1.500 | 0.918 | 0.452 | 0.490 | 0.594 | 0.585 | 0.989 | 0.560 |
| TMDM | 0.742 | 0.642 | 0.535 | 0.474 | 0.514 | 0.460 | 0.411 | 0.409 | 0.275 | 0.314 | 0.212 | 0.327 | 0.711 | 0.415 |
| NsDiff | 0.687 | 0.563 | 0.448 | 0.448 | 0.461 | 0.447 | 0.278 | 0.344 | 0.251 | 0.290 | 0.201 | 0.303 | 0.640 | 0.374 |
| PPM | **0.447** | **0.432** | **0.376** | **0.394** | **0.381** | **0.390** | **0.247** | **0.306** | **0.242** | **0.268** | **0.182** | **0.267** | **0.524** | **0.323** |

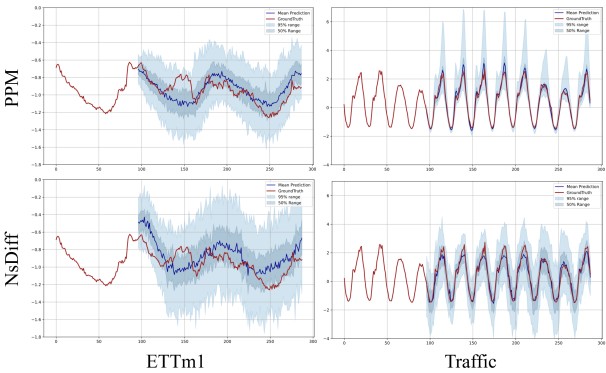

*Figure 3.* Probabilistic prediction interval comparisons on ETTm1 and Traffic datasets with NsDiff.

Second, to assess conditioning quality, we quantify information retention in the learned prior. Finally, we verify how the NLL and Mean MSE objectives complement each other to guide this process. These analyses affirm the synergy between our architectural choices and learning objectives.

### 6.4.1. PUSH-FORWARD MAPPING

Table 4 demonstrates the module's effectiveness in capturing complex distributions in ETTm1 and Traffic datasets. **Gauss.**

*Table 4.* Push-Forward with different prior distribution.

| Dataset | Metric | Gauss. | + PM | Unif. | + PM |
|---|---|---|---|---|---|
| | QICE | 2.286 | **1.580** | 4.280 | **1.626** |
| ETTm2 | CRPS | **0.237** | **0.237** | 0.241 | **0.236** |
| | MSE | 0.250 | **0.247** | 0.250 | **0.246** |
| | MAE | 0.309 | **0.306** | 0.309 | **0.306** |
| | QICE | **2.414** | 2.744 | 3.715 | **2.616** |
| Traffic | CRPS | 0.266 | **0.252** | 0.271 | **0.251** |
| | MSE | 0.542 | **0.524** | 0.543 | **0.520** |
| | MAE | 0.335 | **0.323** | 0.336 | **0.321** |

and **Unif.** mean that the prior distribution is directly used for predictive distribution, while **+PM** indicates adding our push-forward for end-to-end estimation. The results show that applying the push-forward projection consistently improves performance. We further evaluated different priors, such as the Uniform distribution. Notably, even with a prior that substantially deviates from empirical data, our method effectively steers it toward the target distribution. These results confirm the stability and robustness of PPM regarding prior selection. Additional analysis of prior form is provided in Appendix C.2. We also evaluate the effectiveness of the end-to-end design in the Appendix C.3.

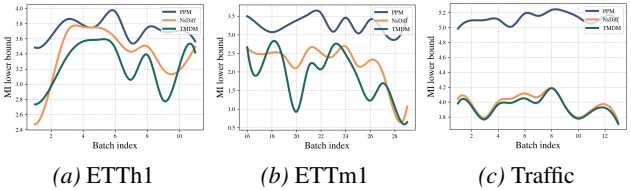

*(a)* ETTh1      *(b)* ETTm1      *(c)* Traffic

*Figure 5.* Mutual-information lower bound between the input $x$ and the prior latent variable $z$ on the test set of ETTh1, ETTm1, and Traffic (batch size=256).

*Table 5.* Ablation studies of the PPM objective function.

| Dataset | ETTm1 | | | Electricity | | |
|---|---|---|---|---|---|---|
| Metric | MSE | CRPS | QICE | MSE | CRPS | QICE |
| w/o-NLL | **0.371** | 0.345 | 6.342 | **0.182** | 0.257 | 8.317 |
| w/o-MM | 0.407 | 0.324 | 1.912 | 0.191 | 0.213 | 5.697 |
| PPM | 0.381 | **0.314** | **1.782** | **0.182** | **0.206** | **2.435** |

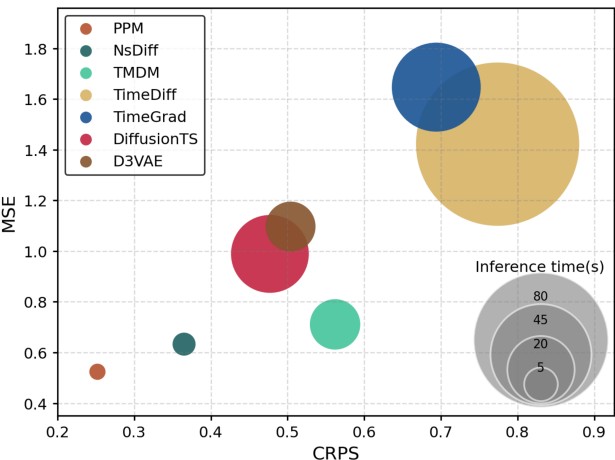

*Figure 6.* Inference time comparison with different models on Traffic dataset, with history window $H = 96$, future length $L = 192$. The number of samples is set to 100.

### 6.4.2. PARAMETRIC PRIOR ANALYSIS

To further substantiate the advantage of our method in parametric prior modeling, we compute the mutual-information lower bound (MI lower bound) (Oord et al., 2018) between the input sequence $x$ and the prior latent variable $z$ on the test sets of ETTh1, ETTm1, and Traffic (batch size = 256) in the Figure 5. The "Batch index" on the x-axis serves as a proxy for the progression of time, corresponding to sequentially unshuffled data windows of test sets. A higher MI lower bound indicates that the prior representation $z$ preserves more information from $x$, which is beneficial for effectively conditioning the subsequent forecasting.

Overall, our method attains a higher and smoother MI lower bound across most batch indices, indicating that the parameterized prior preserves richer and more stable conditional information from the input, providing a more reliable conditioning signal for forecasting. Notably, on Traffic, with complex dynamics, our curve remains stable and maintains a margin over the baselines, suggesting a robust input-aware prior that captures the conditioning information. An ablation study on the latent variable $z$, comparing a fixed identity covariance matrix with a learned parameterized variance, is provided in AppendixC.4.

### 6.4.3. OBJECTIVE FUNCTION

To investigate the contributions of the PPM's objective function, we conduct ablation experiments on ETTm1 and Electricity datasets in Table 5. Specifically, we evaluated the following variants: **w/o-MM**: PPM loss without Mean MSE term; **w/o-NLL**: PPM loss without Negative Log Likelihood term. It can be observed that removing the NLL term leads to performance degradation mainly in CRPS and QICE, highlighting the effectiveness of the NLL term in guiding

the model to learn the predictive distribution. Similarly, removal of the MM loss results in a performance drop mainly in MSE, indicating the importance of MM loss to stabilize training and enhance the ability of point forecasting. We also analyze the impact of different kernel functions on the results in the Appendix C.5.

### 6.5. Inference Complexity

We analyze the inference-time complexity of our framework. Given history $x$, PPM first evaluates the backbone $f_\theta(x)$ and then applies a single-pass map $g_\phi(\cdot)$ to $K$ reparameterized draws to obtain $K$ samples. Hence,

$$T_{\text{PPM}} = O(B \cdot \text{Cost}(f_\theta) + BK \cdot \text{Cost}(g_\phi)).$$

By contrast, diffusion forecasters require a $T$-step denoising chain per sample:

$$T_{\text{Diffusion}} = O(BKT \cdot \text{Cost}(g)).$$

Where $g$ denotes the denoising network. Compared to diffusion, PPM replaces the sequential $T$-step refinement with a single mapping, leading to an approximately $\Theta(T)$ reduction in the dominant inference term, and it parallelizes naturally over the $K$ samples. Figure 6 reports inference-time comparisons on Traffic ($H = 96$, $L = 192$, batch size 1). It can be observed that our model achieves the best point-forecasting and probabilistic modeling performance while delivers a 2× to 100× speedup in inference relative to leading diffusion-based models. However, computing the KDE likelihood (e.g., during training) requires kernel aggregation, which introduces an additional $O(BCLK)$ computational overhead to $T_{\text{PPM}}$. Since this step is typically bypassed at inference, PPM strikes a trade-off between training complexity and inference latency.

# 7. Conclusion

We introduced Parametric Prior Mapping (PPM), a framework that combines the efficiency of parametric methods with the expressiveness of generative modeling. By learning a push-forward mapping from a parametric prior, PPM captures complex non-stationary dynamics while avoiding the overhead of iterative diffusion. Experiments show PPM achieves state-of-the-art accuracy and calibration with up to $100\times$ faster inference, offering a scalable solution for probabilistic time-series forecasting.

**Limitations:** Our method uses KDE to estimate the conditional predictive density, with a bandwidth $h$ that controls the bias–variance trade-off. Although performance is usually stable within a reasonable range, extreme or rapidly shifting regimes can amplify sensitivity to $h$. Future work will explore more robust density estimators and adaptive bandwidth strategies (e.g., learnable or local/multi-bandwidth KDE), as well as training objectives beyond likelihood, such as strictly proper scoring rules (e.g., CRPS), to improve calibration and reduce dependence on KDE tuning.

In addition, the current PPM does not take into account the joint distribution modeling of the labels. Although independent-point learning is an effective strategy for time series forecasting, further advancements in this direction will be explored in the future, for example, by employing the energy score or variogram score as a learning objective.

# Acknowledgements

This work was supported by the National Science Foundation of China (62476289).

# Impact Statement

This paper presents work whose goal is to advance the field of Machine Learning. There are many potential societal consequences of our work, none which we feel must be specifically highlighted here.

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

# A. Additional Theoretical Results for PPM

**Log-domain stabilization via truncation (analysis matches implementation).**  In implementation, we compute $\log \hat{q}_h$ by the log-sum-exp trick and apply a lower truncation:

$$\log \tilde{q}_h(y|\boldsymbol{x}) := \max \big\{ \log \hat{q}_h(y|\boldsymbol{x}), \ \log \varepsilon \big\}, \qquad \varepsilon > 0, \tag{17}$$

equivalently $\tilde{q}_h(y|\boldsymbol{x}) = \max \big\{ \hat{q}_h(y|\boldsymbol{x}), \ \varepsilon \big\}$. We analyze this truncated objective since it coincides with the implemented training loss. The per-example PPM NLL (marginal/composite form) is thus

$$\hat{\mathcal{L}}_{\mathrm{NLL}}^{(i)}(\theta, \phi) := -\frac{1}{CL} \sum_{c,t} \log \tilde{q}_h(y_{c,t}|\boldsymbol{x}). \tag{18}$$

**Gaussian kernel and logsumexp form.**  For the Gaussian kernel $\mathcal{K}(u) = \frac{1}{\sqrt{2\pi}} \exp(-u^2/2)$,

$$\hat{q}_h(y|\boldsymbol{x}) = \frac{1}{\sqrt{2\pi}Kh} \sum_{k=1}^{K} \exp\left(-\frac{(y-\hat{y}^k)^2}{2h^2}\right), \tag{19}$$

and its log-density can be computed stably as

$$\log \hat{q}_h(y|\boldsymbol{x}) = -\log(\sqrt{2\pi}Kh) \ + \ \mathrm{LSE}_{k=1}^{K}\left(-\frac{(y-\hat{y}^k)^2}{2h^2}\right), \tag{20}$$

where $\mathrm{LSE}(a_1, \ldots, a_K) := \log \sum_{k=1}^{K} e^{a_k}$. Finally we apply (17).

## A.1. Consistency and finite-$(K, h)$ error decomposition

**Setup and assumptions.**  We use standard KDE regularity conditions: bounded kernel with finite moments and a twice-differentiable target density. These are summarized below.

**Assumption A.1** (Kernel and smoothness).  The kernel $\mathcal{K} : \mathbb{R} \to \mathbb{R}_+$ satisfies: (i) $\int \mathcal{K}(u) \, du = 1$; (ii) $\int u \, \mathcal{K}(u) \, du = 0$; (iii) $\int u^2 \mathcal{K}(u) \, du =: m_2 < \infty$; (iv) $\|\mathcal{K}\|_\infty \leq \kappa$ and $v_{\mathcal{K}} := \int \mathcal{K}(u)^2 \, du < \infty$. Moreover, for each fixed $\boldsymbol{x}$, $q_{\theta,\phi}(\cdot|\boldsymbol{x})$ is twice continuously differentiable and $\sup_y |\partial_{yy} q_{\theta,\phi}(y|\boldsymbol{x})| \leq M$.

**Definition A.2** (Kernel-smoothed density).  Define the kernel-smoothed model density (convolution) by

$$q_h(y|\boldsymbol{x}) := \mathbb{E}_{\hat{y} \sim q_{\theta,\phi}(\cdot|\boldsymbol{x})} \left[ \frac{1}{h} \mathcal{K}\left(\frac{y-\hat{y}}{h}\right) \right] = \big(q_{\theta,\phi}(\cdot|\boldsymbol{x}) * \mathcal{K}_h\big)(y), \tag{21}$$

where $\mathcal{K}_h(u) = \frac{1}{h}\mathcal{K}(u/h)$. We also define the truncated (floored) *population* version

$$\tilde{q}_{h,\mathrm{pop}}(y|\boldsymbol{x}) := \max\{q_h(y|\boldsymbol{x}), \ \varepsilon\}. \tag{22}$$

**Lemma A.3** (Unbiasedness for the smoothed density).  *For any fixed $(\boldsymbol{x}, y)$ and any $h > 0$, the KDE estimator satisfies $\mathbb{E}\big[\hat{q}_h(y|\boldsymbol{x})\big] = q_h(y|\boldsymbol{x})$, where the expectation is over the $K$ i.i.d. samples $\{\hat{y}^k\}$.*

*Proof.*  Starting from (11),

$$\mathbb{E}[\hat{q}_h(y|\boldsymbol{x})] = \mathbb{E}\left[\frac{1}{Kh} \sum_{k=1}^{K} \mathcal{K}\left(\frac{y-\hat{y}^k}{h}\right)\right] = \frac{1}{Kh} \sum_{k=1}^{K} \mathbb{E}\left[\mathcal{K}\left(\frac{y-\hat{y}^k}{h}\right)\right].$$

Since $\hat{y}^k \overset{i.i.d.}{\sim} q_{\theta,\phi}(\cdot|\boldsymbol{x})$, each term has the same expectation, hence the sum equals $K$ times one term:

$$\mathbb{E}[\hat{q}_h(y|\boldsymbol{x})] = \frac{1}{h}\mathbb{E}_{\hat{y} \sim q_{\theta,\phi}(\cdot|\boldsymbol{x})}\left[\mathcal{K}\left(\frac{y-\hat{y}}{h}\right)\right] = q_h(y|\boldsymbol{x}).$$

$\square$

**Lemma A.4** (Smoothing bias of $q_h$). *Under Assumption A.1, for any fixed $(\boldsymbol{x}, y)$,*

$$\left| q_h(y|\boldsymbol{x}) - q_{\theta,\phi}(y|\boldsymbol{x}) \right| \leq \frac{m_2}{2} M h^2. \tag{23}$$

*Proof.* A standard second-order Taylor expansion of $q_{\theta,\phi}(y-hu|\boldsymbol{x})$ around $y$, combined with $\int \mathcal{K}(u)\,du = 1$, $\int u\mathcal{K}(u)\,du = 0$, and $\sup|\partial_{yy} q_{\theta,\phi}| \leq M$, yields (23). $\qquad\square$

**Lemma A.5** (Concentration around $q_h$). *Under Assumption A.1, for any $\delta \in (0,1)$ and fixed $(\boldsymbol{x}, y)$, with probability at least $1 - \delta$,*

$$\left| \hat{q}_h(y|\boldsymbol{x}) - q_h(y|\boldsymbol{x}) \right| \leq \frac{\kappa}{h}\sqrt{\frac{2\log(2/\delta)}{K}}. \tag{24}$$

*Proof.* Each summand in (11) is bounded in $[0, \kappa/h]$ and i.i.d. Applying Hoeffding's inequality to the average yields (24). $\qquad\square$

**Lemma A.6** (Log perturbation under truncation). *Define $\mathrm{clip}_\varepsilon(a) := \max\{a, \varepsilon\}$ for $a \geq 0$. Then for any $a, b \geq 0$ and $\varepsilon > 0$,*

$$\left| \log(\mathrm{clip}_\varepsilon(a)) - \log(\mathrm{clip}_\varepsilon(b)) \right| \leq \frac{|a - b|}{\varepsilon}. \tag{25}$$

*Proof.* $\mathrm{clip}_\varepsilon$ is 1-Lipschitz, and $\log(\cdot)$ is $(1/\varepsilon)$-Lipschitz on $[\varepsilon, \infty)$. Composing the two yields (25). $\qquad\square$

**Theorem A.7** (Finite-$(K, h)$ NLL error decomposition and consistency (log-truncation)). *Let $\hat{\mathcal{L}}_{\mathrm{NLL}}^{(i)}$ be defined in (18) with truncation $\tilde{q}_h = \max\{\hat{q}_h, \varepsilon\}$. Define the truncated "population" counterparts*

$$\mathcal{L}_{h,\varepsilon}^{(i)}(\theta, \phi) := -\frac{1}{CL}\sum_{c,t} \log\max\{q_h(y_{c,t}|\boldsymbol{x}), \varepsilon\}, \qquad \mathcal{L}_\varepsilon^{(i)}(\theta, \phi) := -\frac{1}{CL}\sum_{c,t}\log\max\{q_{\theta,\phi}(y_{c,t}|\boldsymbol{x}), \varepsilon\}. \tag{26}$$

*Under Assumption A.1, for any $\delta \in (0,1)$, with probability at least $1 - \delta$,*

$$\left| \hat{\mathcal{L}}_{\mathrm{NLL}}^{(i)} - \mathcal{L}_\varepsilon^{(i)} \right| \leq \underbrace{\frac{\kappa}{\varepsilon h}\sqrt{\frac{2\log(2CL/\delta)}{K}}}_{\textit{finite-}K\textit{ stochastic term}} + \underbrace{\frac{m_2 M}{2\varepsilon}h^2}_{\textit{smoothing bias term}}. \tag{27}$$

*Consequently, if $h \to 0$ and $Kh^2 \to \infty$, then $\hat{\mathcal{L}}_{\mathrm{NLL}}^{(i)}(\theta, \phi) \to \mathcal{L}_\varepsilon^{(i)}(\theta, \phi)$ in probability.*

*Proof.* We decompose

$$|\hat{\mathcal{L}}_{\mathrm{NLL}} - \mathcal{L}_\varepsilon| \leq |\hat{\mathcal{L}}_{\mathrm{NLL}} - \mathcal{L}_{h,\varepsilon}| + |\mathcal{L}_{h,\varepsilon} - \mathcal{L}_\varepsilon|.$$

For the finite-$K$ term, apply Lemma A.5 at each of the $CL$ coordinates and use a union bound to ensure probability at least $1 - \delta$ overall. Then apply Lemma A.6 to translate density error into truncated log error. For the smoothing-bias term, apply Lemma A.4 and Lemma A.6 coordinate-wise. Both terms vanish under $h \to 0$ and $Kh^2 \to \infty$. $\qquad\square$

**Corollary A.8** (Rate interpretation). *Equation (27) implies*

$$\left| \hat{\mathcal{L}}_{\mathrm{NLL}}^{(i)} - \mathcal{L}_\varepsilon^{(i)} \right| = O_p\left( \frac{1}{\varepsilon}\left( \frac{1}{h\sqrt{K}} + h^2 \right) \right),$$

*illustrating the trade-off between finite-$K$ fluctuation and smoothing bias.*

### A.2. Expressiveness of push-forward conditional generation.

For each history $\boldsymbol{x}$, PPM generates forecasts by a push-forward map

$$\hat{y} = g_\phi(\boldsymbol{z}), \qquad \boldsymbol{z} \sim p_\theta(\cdot \mid \boldsymbol{x}).$$

Recall that for a measurable map $T : \mathcal{Z} \to \mathcal{Y}$ and a measure $p_{\boldsymbol{z}}$ on $\mathcal{Z}$, the push-forward $T_\# p_{\boldsymbol{z}}$ is defined by $(T_\# p_{\boldsymbol{z}})(A) = p_{\boldsymbol{z}}(T^{-1}(A))$ for any measurable $A \subseteq \mathcal{Y}$.

**Setup.** Fix $\boldsymbol{x}$ and denote the ground-truth conditional distribution on $\mathbb{R}$ by $p^*(\cdot \mid \boldsymbol{x})$. We show that the push-forward family $\{g_{\phi\#}p_\theta(\cdot \mid \boldsymbol{x})\}$ is dense in $W_1$.

**Assumption A.9** (Target regularity). For each $\boldsymbol{x}$, $p^*(\cdot \mid \boldsymbol{x})$ on $\mathbb{R}$ has a continuous CDF and a finite first moment.

**Lemma A.10** (Existence of a conditional transport from the prior). *Under Assumption A.9, for each fixed $\boldsymbol{x}$ there exists a measurable map $T_{\boldsymbol{x}} : \mathbb{R}^D \to \mathbb{R}$ such that*

$$(T_{\boldsymbol{x}})_{\#}p_\theta(\cdot \mid \boldsymbol{x}) \;=\; p^*(\cdot \mid \boldsymbol{x}).$$

*Proof.* Since $\mathbb{R}^D$ is a standard Borel space, there exists a Borel isomorphism $\psi : \mathbb{R}^D \to (0, 1)$ such that if $\boldsymbol{z} \sim p_\theta(\cdot \mid \boldsymbol{x})$ then $U := \psi(\boldsymbol{z}) \sim \mathrm{Unif}(0, 1)$. Let $F_{\boldsymbol{x}}$ be the CDF of $p^*(\cdot \mid \boldsymbol{x})$ and define the generalized inverse $F_{\boldsymbol{x}}^{-1}(u) := \inf\{y \in \mathbb{R} : F_{\boldsymbol{x}}(y) \geq u\}$. Set

$$T_{\boldsymbol{x}}(\boldsymbol{z}) \;:=\; F_{\boldsymbol{x}}^{-1}\big(\psi(\boldsymbol{z})\big).$$

Then $T_{\boldsymbol{x}}(\boldsymbol{z}) = F_{\boldsymbol{x}}^{-1}(U) \sim p^*(\cdot \mid \boldsymbol{x})$, which implies $(T_{\boldsymbol{x}})_{\#}p_\theta(\cdot \mid \boldsymbol{x}) = p^*(\cdot \mid \boldsymbol{x})$ by the definition of push-forward. $\square$

**Lemma A.11** ($W_1$ control by map approximation under a fixed latent). *Let $\boldsymbol{z} \sim p_\theta(\cdot \mid \boldsymbol{x})$ and let $g, T : \mathbb{R}^D \to \mathbb{R}$ be measurable. Then*

$$W_1\big(g_{\#}p_\theta(\cdot \mid \boldsymbol{x}), \; T_{\#}p_\theta(\cdot \mid \boldsymbol{x})\big) \leq \mathbb{E}\big[\,|g(\boldsymbol{z}) - T(\boldsymbol{z})|\,\big]. \tag{28}$$

*Proof.* Define $Y_1 = g(\boldsymbol{z})$ and $Y_2 = T(\boldsymbol{z})$ with the *same* latent $\boldsymbol{z}$. Then $\mathcal{L}(Y_1) = g_{\#}p_\theta(\cdot \mid \boldsymbol{x})$ and $\mathcal{L}(Y_2) = T_{\#}p_\theta(\cdot \mid \boldsymbol{x})$ by the definition of push-forward. The joint law of $(Y_1, Y_2)$ induced by $\boldsymbol{z}$ is a valid coupling of these two marginals, hence

$$W_1(\mathcal{L}(Y_1), \mathcal{L}(Y_2)) \leq \mathbb{E}|Y_1 - Y_2| = \mathbb{E}\big[\,|g(\boldsymbol{z}) - T(\boldsymbol{z})|\,\big],$$

which proves (28). $\square$

**Assumption A.12** (Universal approximation in $L^1(p_\theta(\cdot \mid \boldsymbol{x}))$). For each fixed $\boldsymbol{x}$ and any measurable $T_{\boldsymbol{x}} : \mathbb{R}^D \to \mathbb{R}$ with $\mathbb{E}_{\boldsymbol{z} \sim p_\theta(\cdot|\boldsymbol{x})}|T_{\boldsymbol{x}}(\boldsymbol{z})| < \infty$, the network class $\{g_\phi(\cdot)\}$ is dense in $L^1(p_\theta(\cdot \mid \boldsymbol{x}))$: for any $\eta > 0$ there exists $\phi$ such that

$$\mathbb{E}_{\boldsymbol{z} \sim p_\theta(\cdot|\boldsymbol{x})}\big|g_\phi(\boldsymbol{z}) - T_{\boldsymbol{x}}(\boldsymbol{z})\big| < \eta. \tag{29}$$

**Theorem A.13** (Universality of conditional push-forward under the conditional prior ($W_1$)). *Under Assumptions A.9 and A.12, for any fixed $\boldsymbol{x}$ and any $\epsilon > 0$, there exists $\phi$ such that the generated conditional distribution*

$$q_\phi(\cdot \mid \boldsymbol{x}) \;:=\; (g_\phi)_{\#}p_\theta(\cdot \mid \boldsymbol{x})$$

*satisfies*

$$W_1\big(q_\phi(\cdot \mid \boldsymbol{x}), \; p^*(\cdot \mid \boldsymbol{x})\big) < \epsilon. \tag{30}$$

*Proof.* By Lemma A.10, there exists $T_{\boldsymbol{x}}$ such that $p^*(\cdot \mid \boldsymbol{x}) = (T_{\boldsymbol{x}})_{\#}p_\theta(\cdot \mid \boldsymbol{x})$. By Assumption A.12, choose $\phi$ so that $\mathbb{E}_{\boldsymbol{z} \sim p_\theta(\cdot|\boldsymbol{x})}|g_\phi(\boldsymbol{z}) - T_{\boldsymbol{x}}(\boldsymbol{z})| < \epsilon$. Applying Lemma A.11 with $g = g_\phi$ and $T = T_{\boldsymbol{x}}$ yields (30). $\square$

**Corollary A.14** (Weak convergence). *Since $W_1$ metrizes weak convergence plus first-moment control on $\mathbb{R}$, Theorem A.13 implies $q_\phi(\cdot \mid \boldsymbol{x}) \Rightarrow p^*(\cdot \mid \boldsymbol{x})$ as $\epsilon \to 0$.*

*Proof.* Full proof in Appendix A.2.

### A.3. Optimization properties (gradient structure and MM stabilization)

**Setup.** We derive the exact gradient of the log-truncated KDE-NLL with respect to the samples and show how MM provides a dense first-moment anchor when responsibilities become highly concentrated.

**Gaussian kernel specialization.** Define $s_k(y) := \exp\left(-\frac{(y-\hat{y}^k)^2}{2h^2}\right)$. Then (20) implies that (before truncation) the responsibilities satisfy

$$\bar{\omega}_j(y) := \frac{s_j(y)}{\sum_{k=1}^K s_k(y)}, \qquad \sum_{j=1}^K \bar{\omega}_j(y) = 1. \tag{31}$$

**Lemma A.15** (Exact gradient of log-truncated KDE-NLL w.r.t. samples). *Let $\ell_\varepsilon(y) := -\log \tilde{q}_h(y|\boldsymbol{x})$ with $\tilde{q}_h = \max\{\hat{q}_h, \varepsilon\}$. Then for any $j \in \{1, \ldots, K\}$,*

$$\frac{\partial \ell_\varepsilon(y)}{\partial \hat{y}^j} = \mathbf{1}\{\hat{q}_h(y|\boldsymbol{x}) \geq \varepsilon\} \cdot \frac{1}{h^2} \bar{\omega}_j(y)(\hat{y}^j - y). \tag{32}$$

*Proof.* If $\hat{q}_h(y) \geq \varepsilon$, then $\tilde{q}_h(y) = \hat{q}_h(y)$ and $\ell_\varepsilon(y) = -\log \hat{q}_h(y)$. Differentiating the Gaussian KDE (equivalently, differentiating (20)) yields $\frac{\partial}{\partial \hat{y}^j}[-\log \hat{q}_h(y)] = \frac{1}{h^2}\bar{\omega}_j(y)(\hat{y}^j - y)$. If $\hat{q}_h(y) < \varepsilon$, then $\tilde{q}_h(y) = \varepsilon$ is constant in $\{\hat{y}^k\}$, hence the gradient is 0. $\square$

**Corollary A.16** (Sparse-gradient regime and truncation effect). *If $h$ is small (or samples are dispersed), then $s_k(y)$ is sharply peaked around samples closest to $y$, causing $\bar{\omega}_j(y)$ to concentrate on a small subset of indices, and the gradient becomes effectively winner-take-all. Moreover, when $\hat{q}_h(y)$ falls below the floor $\varepsilon$, truncation sets the gradient to zero, preventing numerically unstable updates induced by extremely small likelihood values.*

**Mean MSE (MM) anchor.** Let $\bar{y} := \frac{1}{K}\sum_{k=1}^K \hat{y}^k$ denote the sample mean at a coordinate. The per-coordinate MM term is $\ell_{\mathrm{MM}}(y) := (y - \bar{y})^2$.

**Lemma A.17** (Gradient of MM is dense across samples). *For any $j \in \{1, \ldots, K\}$,*

$$\frac{\partial}{\partial \hat{y}^j}(y - \bar{y})^2 = \frac{2}{K}(\bar{y} - y). \tag{33}$$

*Thus the MM gradient is shared equally among all $K$ samples and does not vanish due to responsibility concentration.*

*Proof.* Use $\partial \bar{y}/\partial \hat{y}^j = 1/K$ and differentiate $(y - \bar{y})^2$. $\square$

**Theorem A.18** (Combined gradient and stabilization mechanism). *Consider the combined per-coordinate loss*

$$\ell(y) := \alpha \cdot \left[-\log \tilde{q}_h(y|\boldsymbol{x})\right] + (y - \bar{y})^2, \tag{34}$$

*where $\tilde{q}_h = \max\{\hat{q}_h, \varepsilon\}$. Then for each sample $j$,*

$$\frac{\partial \ell(y)}{\partial \hat{y}^j} = \alpha \cdot \mathbf{1}\{\hat{q}_h(y|\boldsymbol{x}) \geq \varepsilon\} \cdot \frac{1}{h^2} \bar{\omega}_j(y)(\hat{y}^j - y) + \frac{2}{K}(\bar{y} - y). \tag{35}$$

*In the sparse-gradient regime (Corollary A.16), the MM term provides a dense, low-variance direction that anchors the sample mean toward $y$ and mitigates early-stage instability caused by finite $(K, h)$.*

*Proof.* Sum the gradients from Lemma A.15 and Lemma A.17 with the weight $\alpha$. $\square$

**Corollary A.19** (Practical implication: complementary roles of NLL and MM). *Equation (35) shows that KDE-NLL shapes distributional structure through responsibility-weighted residuals (when $\hat{q}_h \geq \varepsilon$), while MM controls the first moment (sample mean) with dense gradients. This complementarity is most beneficial when finite-$K$ and small-$h$ cause responsibility concentration.*

## B. Experiments Details

### B.1. Datasets Details

We evaluate the performance of our proposed PPM on seven popular real-world time series datasets. (1) **Electricity transformer temperature (ETT)** (Wu et al., 2021) contains the power load features and oil temperature collected from electricity transformers, consisting of seven features. Following the same protocol as Informer (Zhou et al.,

2021), we split the data into four datasets: ETTh1, ETTh2, ETTm1, and ETTm2. (2) **Electricity Consuming Load (ECL)** (Nie et al., 2022) data contains the hourly electricity consumption of 321 customers from 2012 to 2014. (3) **Traffic** (Wu et al., 2021; Li et al., 2025a) data is a collection of hourly data from California Department of Transportation and describes the occupancy rate of different lanes measured by different sensors on the San Francisco highway. (4) **Weather** (Nie et al., 2022) data is recorded, 21 meteorological indicators collected every 10 minutes from the Weather Station of the Max Planck Biogeochemistry Institute in 2020. We preprocess all datasets following (Wu et al., 2021) and normalize them with the z-score normalization. Due to memory constraints, for multivariate datasets with a large number of channels (e.g., Traffic, Electricity, and Weather), we set the sliding stride of the sampling window to the prediction length. This reduces the number of evaluated windows and the memory footprint, while still covering the entire test horizon so that all test data are included in inference and evaluation. All baselines use this setting during evaluation.

### B.2. Baselines

We select several strong forecasting baselines for comparison, including probabilistic forecasting methods: DeepAR (Salinas et al., 2020), TimeGrad (Rasul et al., 2021), D3VAE (Li et al., 2022), TimeDiff (Shen & Kwok, 2023), DiffusionTs (Yuan & Qiao, 2024), TMDM (Li et al., 2024), and NsDiff (Ye et al., 2025).

The descriptions of the baseline methods are presented as follows:

1. DeepAR leverages an autoregressive recurrent neural network (RNN) to predict the parameters of a probability distribution, enabling probabilistic forecasting at each time step.

2. TimeGrad combines the autoregressive method and a diffusion probabilistic model to model high-dimensional distributions at each time step.

3. D3VAE incorporates a bidirectional Variational Autoencoder and component separation to achieve Temporal Data Augmentation for diffusion generation.

4. TimeDiff leverages futuremixup and autoregressive initialization for a non-autoregressive diffusion model to predict.

5. DiffusionTs generates multivariate time series using an encoder-decoder transformer with disentangled temporal representations. It reconstructs the sample directly in each diffusion step and incorporates a Fourier-based loss term for improved quality.

6. TMDM integrates conditional information into the diffusion forward process with a Transformer model to enhance the estimation of uncertainty distributions over a sequence.

7. NsDiff additionally incorporates an explicit variance estimation module to parameterize the prior uncertainty, trained under supervision from variances computed over sliding-window segments.

### B.3. Metrics

**CRPS (Continuous Ranked Probability Score):** The continuous ranked probability score (CRPS) (Matheson & Winkler, 1976) measures the compatibility of a cumulative distribution function (CDF) $F$ with an observation $x$ as

$$\text{CRPS}(F, x) = \int_{\mathbb{R}} \left( F(z) - \mathbb{I}\{x \leq z\} \right)^2 dz$$

where $\mathbb{I}\{z \leq x\}$ is an indicator function that equals one if $r \leq x$ and zero otherwise. As a proper scoring function, CRPS achieves its minimum when the predictive distribution F matches the true data distribution. Empirical CDF $\hat{F}(z) = \frac{1}{n} \sum_{i=1}^{n} \mathbb{I}\{x_i \leq z\}$ is employed as a natural approximation of the predictive CDF. We generated 100 samples to approximate the distribution.

**QICE (Quantile Interval Calibration Error):** The quantile interval calibration error (QICE) (Han et al., 2022) quantifies the deviation between the proportion of true data contained within each quantile interval (QI) and the optimal proportion, which is $1/M$ for all intervals. To compute QICE, we divide the generated $y$-samples into $M$ quantile intervals with roughly equal sizes, corresponding to the boundaries of the estimated quantiles. Under the optimal scenario, when the learned

distribution matches the true distribution, each QI should contain approximately $1/M$ of the true data. QICE is formally defined as the mean absolute error between the observed and optimal proportions, and can be expressed as:

$$\text{QICE} := \frac{1}{M} \sum_{m=1}^{M} \left| r_m - \frac{1}{M} \right|$$

where $r_m = \frac{1}{N} \sum_{n=1}^{N} \mathbb{I}_{y_n \geq \hat{y}_n^{low m}} \cdot \mathbb{I}_{y_n \geq \hat{y}_n^{high m}}$. Here, $\mathbb{I}_{\text{condition}}$ is an indicator function. The terms $\hat{y}_n^{low m}$ and $\hat{y}_n^{high m}$ denote the lower and upper boundaries of the $m$-th quantile interval, respectively. Intuitively, under ideal conditions with sufficient samples, QICE should approach 0, indicating that each QI contains the expected proportion of data.

*Table 6.* Hyperparameters of PPM.

| Dataset | Batch size | Learning rate | DatasetSplit | Hidden Dim $H$ |
|---|---|---|---|---|
| ETTh1 | 64 | 0.0001 | (6:2:2) | 256 |
| ETTh2 | 64 | 0.0001 | (6:2:2) | 256 |
| ETTm1 | 128 | 0.0001 | (6:2:2) | 256 |
| ETTm2 | 128 | 0.0001 | (6:2:2) | 256 |
| Electricity | 16 | 0.0005 | (7:1:2) | 256 |
| Traffic | 16 | 0.001 | (7:1:2) | 256 |
| Weather | 128 | 0.0001 | (7:1:2) | 512 |

### B.4. Implementation Details

The proposed model is trained using the Adam optimizer. Early stopping is applied after 5 epochs without improvement, with a maximum of 30 training epochs. The parameters of the conditional network are set to the default parameters of the original paper. More details of the model are shown in the Table.6. All experiments are implemented using PyTorch and executed on an NVIDIA RTX A100 40GB GPU. The sensitivity analysis could be found in the Appendix C.6.

## C. Experiments

### C.1. Conditional Network

*Table 7.* Performance promotion by applying the different conditional point forecasting models on our framework. Lower MSE/MAE/CRPS/ indicates better point accuracy, distribution quality, and calibration.

| Dataset | ETTh1 | | | Electricity | | | Traffic | | |
|---|---|---|---|---|---|---|---|---|---|
| **Method** | MSE | MAE | CRPS | MSE | MAE | CRPS | MSE | MAE | CRPS |
| ns_Transformer | 0.592 | 0.546 | - | 0.203 | 0.304 | - | 0.633 | 0.350 | - |
| ns_Transformer(PPM) | 0.544 | 0.504 | 0.370 | 0.209 | 0.309 | 0.234 | 0.655 | 0.359 | 0.281 |
| MLP | 0.517 | 0.483 | - | 0.196 | 0.276 | - | 0.610 | 0.364 | - |
| MLP(PPM) | 0.447 | 0.432 | 0.337 | 0.182 | 0.267 | 0.206 | 0.524 | 0.323 | 0.252 |
| iTransformer | 0.438 | 0.432 | - | 0.159 | 0.251 | - | 0.453 | 0.291 | - |
| iTransformer(PPM) | 0.443 | 0.434 | 0.332 | 0.164 | 0.255 | 0.199 | 0.464 | 0.299 | 0.231 |

Our PPM framework is designed as a plug-and-play solution that can be seamlessly integrated into existing point forecasting models. One conditional model with stronger point forecasting capability can provide more accurate conditional information for probabilistic forecasting, thereby effectively enhancing the overall forecasting performance. The conditioning network is replaced with Nsformer (Liu et al., 2022), MLP (Zeng et al., 2023), and iTransformer (Liu et al., 2024), respectively. As shown in Tab.7, the backbone with powerful point forecasting ability could improve the performance of PPM. Meanwhile, PPM can achieve performance comparable to, or even surpassing, that of point forecasting models.

*Table 8.* Push-Forward with different prior forms. Lower MSE/MAE/CRPS/ indicates better point accuracy, distribution quality, and calibration. The **best** results are highlighted in **bold**.

| Dataset | Metric | Gauss. | +PF | Unif. | +PF | Lap. | +PF | Stud.-t | +PF | Logis. | +PF | Gum. | +PF |
|---------|--------|--------|-----|-------|-----|------|-----|---------|-----|--------|-----|------|-----|
| ETTh1 | CRPS | 0.350 | **0.337** | 0.352 | **0.337** | 0.343 | **0.335** | 0.336 | **0.335** | 0.335 | **0.335** | 0.359 | **0.334** |
| | MSE | 0.481 | **0.447** | 0.466 | **0.447** | 0.461 | **0.449** | 0.464 | **0.449** | 0.463 | **0.450** | 0.476 | **0.446** |
| | MAE | 0.459 | **0.432** | 0.448 | **0.432** | 0.444 | **0.434** | 0.446 | **0.434** | 0.446 | **0.435** | 0.454 | **0.432** |
| ETTm2 | CRPS | **0.237** | **0.237** | 0.241 | **0.236** | 0.236 | **0.233** | 0.236 | **0.233** | 0.237 | **0.232** | 0.240 | **0.234** |
| | MSE | 0.250 | **0.247** | 0.250 | **0.246** | 0.250 | **0.244** | 0.250 | **0.243** | 0.250 | **0.244** | 0.251 | **0.243** |
| | MAE | 0.309 | **0.306** | 0.309 | **0.306** | 0.309 | **0.302** | 0.309 | **0.302** | 0.309 | **0.302** | 0.310 | **0.302** |
| Traffic | CRPS | 0.266 | **0.252** | 0.271 | **0.251** | 0.265 | **0.253** | 0.265 | **0.252** | 0.265 | **0.252** | 0.268 | **0.251** |
| | MSE | 0.542 | **0.524** | 0.543 | **0.520** | 0.543 | **0.523** | 0.543 | **0.525** | 0.542 | **0.523** | 0.538 | **0.523** |
| | MAE | 0.335 | **0.323** | 0.336 | **0.321** | 0.336 | **0.323** | 0.335 | **0.323** | 0.335 | **0.322** | 0.332 | **0.321** |

## C.2. Different Form of Prior

In this section, we evaluate more distribution forms suitable for reparameterized sampling for the prior, including Gaussian (Gauss.), Uniform (Unif.), Laplace (Lap.), Student-t (Stud.-t), Logistic (Logis.), and Gumbel (Gum.). Although different prior distributions exhibit distinct statistical properties and thus approximate the true conditional distribution to varying degrees (e.g., Student-t better captures heavy tails and outliers, while Laplace tends to produce sharper peaks with heavier tails), in PPM, the prior mainly serves as a reparameterizable starting point in latent space. Through the learned push-forward mapping that transports the prior to the predictive space conditioned on the input history, the model can adaptively reshape and reconstruct the distribution, thereby reducing sensitivity to the prior family and better matching the complex, time-varying structure of the real data distribution. As shown in the Table 8, the ability of models to capture the complex distribution is enhanced for all prior distributions.

## C.3. End-to-end Strategy

*Table 9.* Effectiveness of end-to-end (E2E) training. Lower MSE/CRPS/QICE indicates better point accuracy, distribution quality, and calibration. The **best** results are highlighted in **bold**.

| Dataset | ETTh1 | | | ETTm2 | | | Traffic | | |
|---------|-------|------|------|-------|------|------|---------|------|------|
| **method** | MSE | CRPS | QICE | MSE | CRPS | QICE | MSE | CRPS | QICE |
| Gauss. | 0.449 | 0.356 | 4.468 | 0.248 | 0.246 | 3.229 | 0.532 | 0.259 | **2.413** |
| + E2E | **0.447** | **0.337** | **1.481** | **0.247** | **0.237** | **1.580** | **0.524** | **0.252** | 2.744 |
| Unif. | 0.448 | 0.357 | 4.472 | 0.250 | 0.241 | 4.280 | 0.543 | 0.271 | 3.715 |
| + E2E | **0.447** | **0.337** | **1.483** | **0.246** | **0.236** | **1.626** | **0.520** | **0.251** | **2.616** |

Effectiveness of End-to-End Training. To empirically validate the superiority of our end-to-end training paradigm, we conducted a comparative analysis on the ETTh1, ETTm2, and Traffic datasets, as summarized in Table 9. We constructed a decoupled two-stage baseline for comparison, employing standard Gaussian and Uniform distributions as priors. In this baseline setting, the parameterized prior is first pre-trained to fit the latent structure; subsequently, the encoder is frozen, and only the final mapping layers are optimized.

In contrast, our method performs joint optimization of the prior and the mapping. The results indicate that the end-to-end approach yields significantly better performance across all datasets. This advantage is particularly pronounced in the QICE metric, suggesting that joint training avoids information loss inherent in decoupled stages. By allowing the learning objective to guide the prior's formation, the end-to-end strategy ensures that the prior distribution explicitly encodes richer conditional information favorable to the prediction target.

## C.4. Latent Prior Analysis on Variance

Table 10 presents a comparison between our PPM and a fixed standard variance setting $\sigma = I$. The results clearly show that degenerating the learned prior into a fixed identity matrix leads to performance degradation across all datasets, indicating that the prior in PPM captures meaningful distributional information.

*Table 10.* Comparison between PPM and fixed covariance setting.

| Method | ETTh1 | | | ETTh2 | | | Electricity | | |
|---|---|---|---|---|---|---|---|---|---|
| Metric | MSE | MAE | CRPS | MSE | MAE | CRPS | MSE | MAE | CRPS |
| PPM | **0.447** | **0.432** | **0.337** | **0.376** | **0.394** | **0.306** | **0.182** | **0.267** | **0.206** |
| $\sigma = I$ | 0.468 | 0.437 | 0.350 | 0.384 | 0.399 | 0.334 | 0.187 | 0.269 | 0.212 |

### C.5. Kernel Function Estimation

*Table 11.* Performance with different kernel functions in three datasets. Lower MSE/CRPS/QICE indicates better point accuracy, distribution quality, and calibration.

| Dataset | ETTh1 | | | ETTm1 | | | Traffic | | |
|---|---|---|---|---|---|---|---|---|---|
| kernel | MSE | CRPS | QICE | MSE | CRPS | QICE | MSE | CRPS | QICE |
| Gauss. | 0.447 | 0.337 | 1.481 | 0.381 | 0.314 | 1.782 | 0.524 | 0.252 | 2.743 |
| Stud.-t | 0.445 | 0.334 | 1.313 | 0.376 | 0.305 | 1.623 | 0.520 | 0.252 | 2.263 |
| Lap. | 0.445 | 0.334 | 1.429 | 0.376 | 0.305 | 1.773 | 0.519 | 0.256 | 3.567 |
| Logis. | 0.443 | 0.336 | 2.604 | 0.373 | 0.306 | 2.763 | 0.521 | 0.262 | 3.549 |
| Cauchy. | 0.443 | 0.334 | 2.005 | 0.374 | 0.305 | 2.247 | 0.520 | 0.253 | 2.577 |

We compares probabilistic forecasting performance across five KDE kernels: Gaussian (Gauss.), Student-t (Stud.-t), Laplace (Lap.), Logistic (Logis.), and Cauchy (Cauchy.) on three datasets (ETTh1, ETTm1, and Traffic), evaluated by MSE, CRPS, and QICE (lower is typically better). As shown in the Table 11, QICE varies much more than MSE/CRPS, indicating that changing the kernel mainly reshapes the predictive density (especially its tail mass and quantile coverage) while having a limited effect on point errors. Student-t achieves consistently lower QICE across all three datasets because its heavier-tailed form better accommodates time-varying and occasional extreme fluctuations in non-stationary series, leading to improved calibration of prediction intervals/quantiles.

### C.6. Sensitivity Analysis

#### C.6.1. SAMPLING COUNT $K$

To evaluate the sensitivity of the parameters $K$, we present the QICE and CRPS results on the ETTh1, ETTm1, Traffic and Weather datasets in Figure 7 with changing the sampling count $K$. The dashed vertical line in the figure indicates the value used in the main experiments. As shown in the figure, increasing the sampling count $K$ from small values leads to a clear improvement in both probabilistic accuracy (CRPS) and calibration (QICE). This is mainly because, with small $K$, the Monte Carlo approximation underlying KDE is coarse and the sample set provides insufficient coverage of the distribution support, resulting in high-variance density estimates. Increasing $K$ improves support coverage and stabilizes the KDE likelihood estimate, thereby substantially enhancing distribution quality and calibration. With further increasing $K$, performance becomes stable and may slightly degrade. This indicates diminishing returns: beyond a moderate $K$, the KDE estimate is already sufficiently stable and additional samples contribute little new information; moreover, under fixed or weakly adaptive bandwidth settings, overly large $K$ can introduce mild over-smoothing or over-dispersion effects, leading to small metric fluctuations. Overall, PPM remains stable across a wide range of $K$, suggesting that the method is not strongly sensitive to this hyperparameter and can maintain reliable performance in practice.

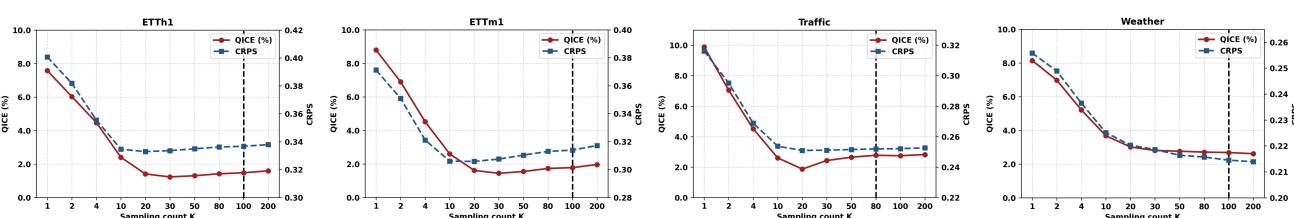

*Figure 7.* Analysis for sampling count K.

### C.6.2. WEIGHT COEFFICIENT $\alpha$

We also present the QICE and CRPS results on the ETTh1, ETTm1, Traffic and Weather datasets in Figure8 with changing the Weight coefficient $\alpha$. The dashed vertical line in the figure indicates the value used in the main experiments. Overall, both QICE and CRPS remain nearly flat across a wide range $\alpha \in [0.05, 2.0]$, with only minor fluctuations on all four datasets, indicating that PPM is not sensitive to the choice of $\alpha$. This suggests that the NLL (distribution learning) term and the MM (mean-anchoring) term provide a stable complement during training: moderate changes in $\alpha$ lead only to limited trade-offs rather than substantial changes in the resulting predictive distribution.

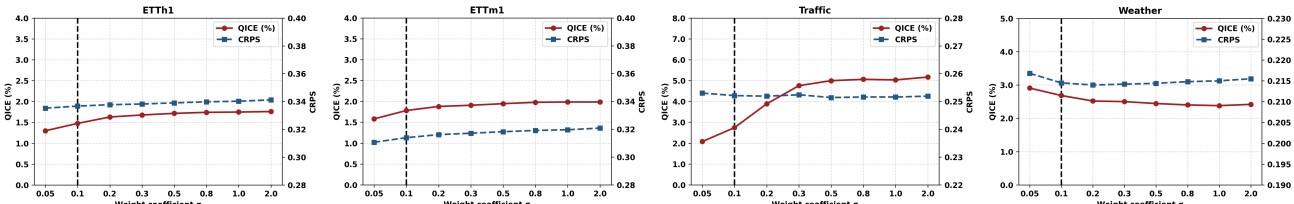

*Figure 8.* Analysis for weight coefficient $\alpha$.

### C.6.3. BANDWIDTH $h$

As shown in Figure 9, we further investigate the effect of the KDE bandwidth $h$ in four datasets. As $h$ increases from small values, both QICE and CRPS drop substantially across all datasets, indicating that a moderate bandwidth alleviates the overly spiky density estimates under small $h$ and yields smoother, more stable likelihood estimation and better probabilistic forecasts (Dehnad, 1987). When $h$ further increases beyond a certain range, the metrics rise again and may degrade noticeably. This is because an overly large bandwidth causes oversmoothing (Wand & Jones, 1994), which washes out fine-grained distributional structure and inflates uncertainty in an uninformative way, thereby worsening calibration and CRPS. Overall, the results suggest that performance is relatively sensitive to the choice of h. To avoid dataset-specific hyperparameter tuning, the value indicated by the dashed line is used in the main experiments, as it provides a reasonable and stable trade-off across datasets.

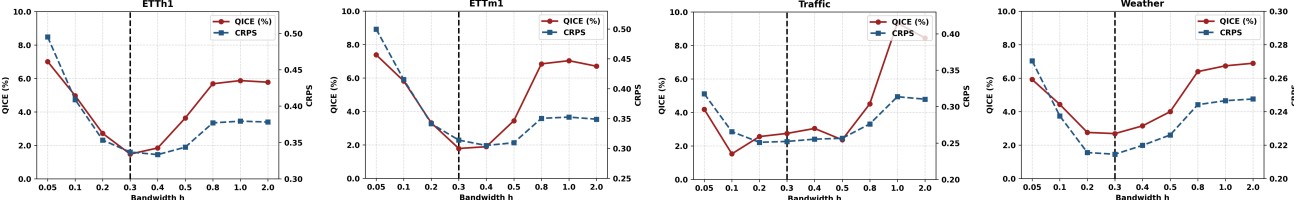

*Figure 9.* Analysis for bandwidth h.

## D. Case Study

To demonstrate the superiority of the proposed method, we visualize the ground truth and predictions of time series across five datasets in Figure 10 and Figure 11.

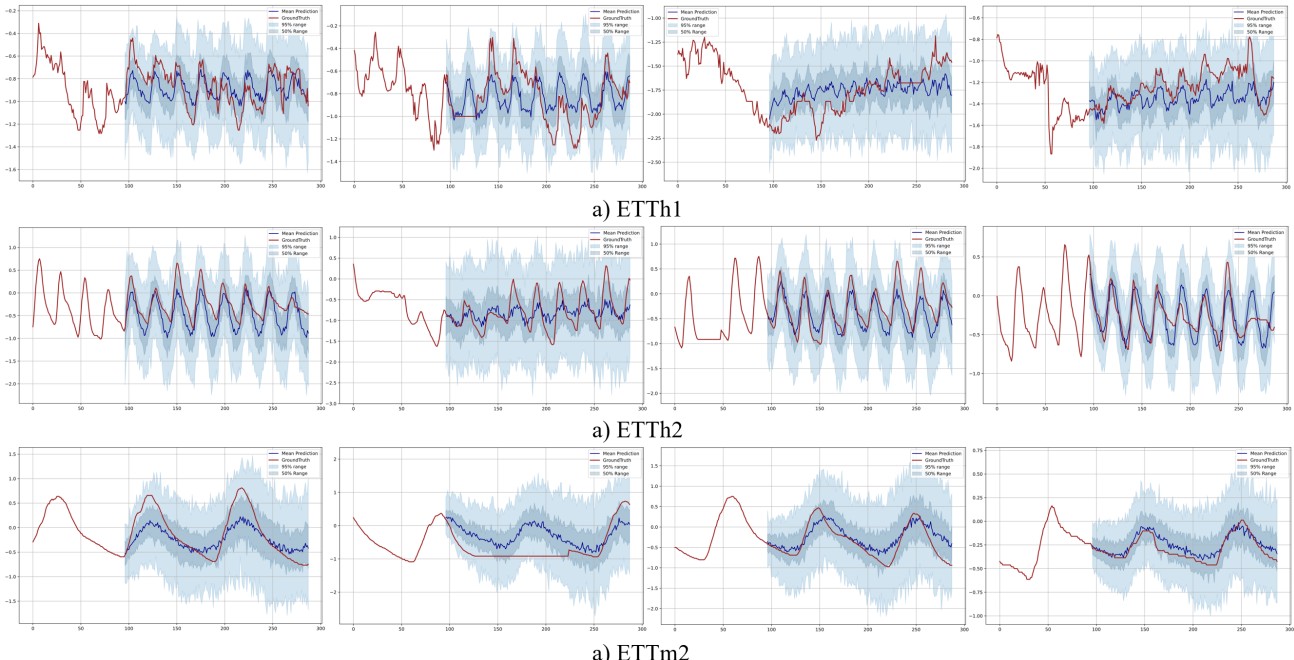

*Figure 10.* Visualization of the ETT datasets.

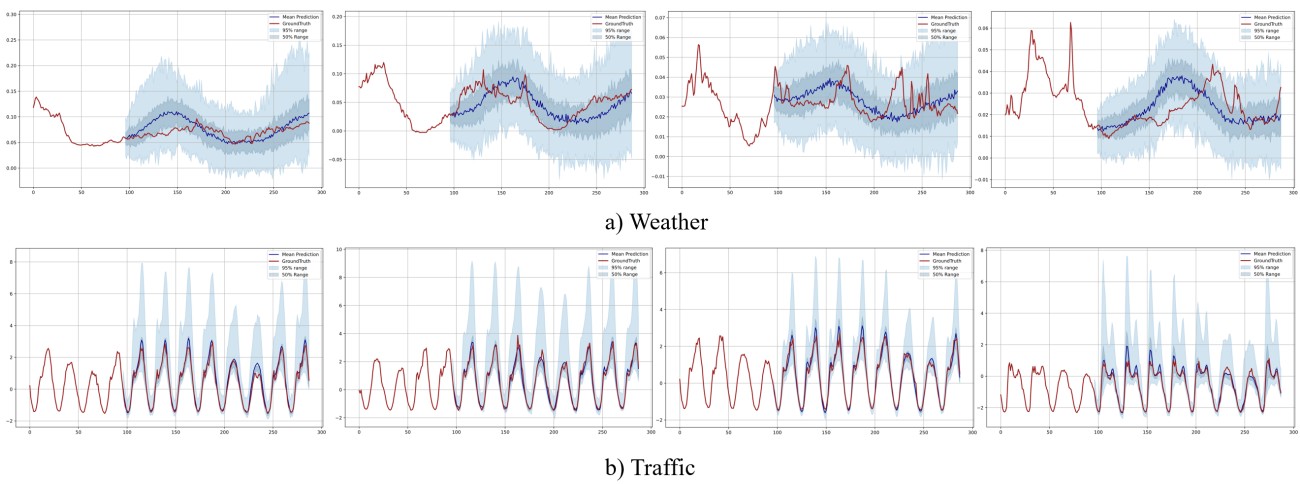

*Figure 11.* Visualization of the Weather and Traffic datasets.

