# OpenReview forum: "Parametric Prior Mapping Framework for Non-stationary Probabilistic Time Series Forecasting"
_ICML.cc/2026/Conference — ICML 2026 regular_

### Official Review · Reviewer_PZr9 · 2026-02-20

**Soundness:** 3
**Presentation:** 4
**Significance:** 2
**Originality:** 2
**Overall Recommendation:** 3
**Confidence:** 4

**Summary:**

This paper proposes an MLE-based framework built upon a latent variable model, in which the variance of the prior distribution is parameterized via a temporal encoder. By doing so, as a main contribution, the proposed model can capture time-varying aleatoric uncertainty. To enable non-parametric likelihood estimation, the authors employ KDE. Additionally, an auxiliary MSE loss is introduced to stabilize gradient updates. To further substantiate the rationale of the proposed method, the paper leverages theoretical properties of KDE and the push-forward measure. Empirically, the proposed method demonstrates strong performance in both probabilistic forecasting and point estimation tasks.

**Compliance With Llm Reviewing Policy:**

Affirmed.

**Key Questions For Authors:**

1. As I understand it, prior approaches such as TMDM do not explicitly model the variance of the latent end-point distribution (i.e., only the mean parameter is modeled via a neural network), whereas the proposed method addresses this limitation by parameterizing the prior variance through a temporal encoder to capture time-varying aleatoric uncertainty. This appears to constitute the main technical contribution as described under "Technical Innovation". If so, it would be important to empirically demonstrate the benefit of modeling the prior variance via a neural network. For example, an ablation study comparing a fixed identity covariance matrix against a learned, neural network-parameterized variance would help substantiate the claimed contribution and clarify its practical significance.

2. The weight of the NLL loss, denoted by $\alpha$, is set to 0.1 throughout all experiments, which is substantially smaller than the weight of the MSE term (set to 1). It would be helpful if the authors could explain why strong performance is achieved even when the NLL component is assigned a relatively small weight.

3. The computation of the NLL loss (Equation (8)) requires sampling from the latent variable, and the paper states that 100 samples are used during training. While Section 6.5 compares inference costs, the computational cost of training relative to baselines is not clearly discussed. It would be valuable to provide a comparison of training time and/or computational complexity. If the proposed method incurs a higher training cost than baselines, this should be explicitly acknowledged as a trade-off.

4. In Section 6.4.1 (lines 376-377), the experimental setting is somewhat unclear. Does "Gauss." and "Unif." indicate that forecasting of $y$ is performed using Gaussian and uniform distributions parameterized solely by $f_\theta$? A more explicit clarification of this setup would improve readability.

5. Please also refer to the comments raised in the weaknesses section above.

**Limitations:**

yes

**Strengths And Weaknesses:**

### Strengths
1. The manuscript is well written and easy to follow. The overall structure is well organized, including the appendix.

2. Most of the necessary experiments have been carefully conducted and clearly presented.

### Weaknesses
1. While it is both necessary and encouraging that the authors attempt to provide a theoretical rationale for the proposed method, Theorems 5.1 and 5.2 appear to have a relatively weak connection to the core task of time series probabilistic forecasting (because for given any $x$, the typical time-series dataset only provide a single conditional $y$), especially when compared to "Gradient Structure and Stabilization" subsection. Their necessity and practical relevance could be more clearly articulated.

2. The distinction between the proposed approach and conventional encoder-decoder-based latent variable models is not sufficiently clear. In the current formulation, $f_\theta$ and $g_\phi$ can be interpreted as playing the roles of encoder and decoder, respectively (particularly if the decoder is assumed to correspond to a Dirac measure). Therefore, it would be helpful to more explicitly clarify how their framework differs from, or extends beyond, standard latent variable models with encoder-decoder architectures.

---

> ### Author Rebuttal · Authors · 2026-03-31
>
> **W.1 The necessity of Theorems 5.1 and 5.2:**
>
> **A.1**: In time-series forecasting, each conditioning input x typically has only one observed target y. **Our method is designed for exactly this setting: we draw K samples from the learned conditional prior and map them through the decoder to obtain multiple predictive samples $\hat{y}^{(i)}$, which are then used to construct a KDE-based approximation of the conditional likelihood.** Therefore, Theorem 5.1 justifies the validity of this finite-K likelihood approximation, rather than assuming repeated observations for a fixed x. Theorem 5.2 complements this by showing that the proposed push-forward family is sufficiently expressive to approximate the true conditional distribution. Together, these results provide theoretical support for both the training objective and the distributional expressiveness of our model.
>
> **W.2 The distinction between PPM and encoder-decoder-based latent variable models:**
>
> **A.2**: Conventional latent-variable models, such as VAEs, typically use an encoder to model an approximate posterior $q(z| y,x)$, thereby explicitly establishing the relationship between the target and the latent variable before using a decoder to model $p(y| z,x)$. In contrast, our method does not require posterior inference. Instead, it directly learns a conditional prior $q(z| x)$ from the input condition and uses a decoder to model $p(y| z,x)$. Moreover, by combining K-sample resampling with kernel density estimation, our framework enables direct maximum-likelihood training on the one observed target. By removing posterior modeling, the framework allows the latent prior to be learned directly from the conditioning context, which is beneficial for capturing dynamic uncertainty in non-stationary time series.
>
> **Q.1 Analysis for prior variance:**
>
> **A.3**: Learning the parameterized prior directly from the encoder is indeed an important part of our contribution. We have conducted the MI lower bound between the input and the prior latent variable in the paper, which shows that PPM can effectively capture conditional information.
>
> To further substantiate this point, we provide additional statistical analysis. We show the mean and standard deviation of the learned prior variance on the test sets across different datasets, together with its average KL divergence from the standard Gaussian distribution. **The results show that the learned variance is clearly different from the identity covariance.**
>
> We conduct an extra ablation by replacing the learned variance with an identity covariance matrix after training and comparing the results with our full method. **These results show that replacing the learned covariance leads to a clear performance drop.**
>
> | Dataset  | ETTh1 | ETTh2 | Elec |
> |:--------:|:-----:|:-----:|:-----------:|
> | var_mean | 2.356 | 2.298 |    0.310    |
> | var_std  | 0.542 | 0.884 |    1.153    |
> | KL_mean  | 1.764 | 1.878 |    2.500    |
>
> | Dataset  | ETTh1 | ETTh1 | ETTh1 | ETTh2 | ETTh2 | ETTh2 | Elec | Elec | Elec |
> |:--------:|:-----:|:-----:|:-----:|:-----:|:-----:|:-----:|:-----------:|:-----------:|:-----------:|
> | Metric   |  MSE  |  MAE  | CRPS  |  MSE  |  MAE  | CRPS  |     MSE     |     MAE     |    CRPS     |
> | sigma=1  | 0.468 | 0.437 | 0.350 | 0.384 | 0.399 | 0.334 |    0.187    |    0.269    |    0.212    |
> | PPM      | 0.447 | 0.432 | 0.337 | 0.376 | 0.394 | 0.306 |    0.182    |    0.267    |    0.206    |
>
> **Q.2 : The NLL weight 0.1 is small.**
>
> **A.4**: Although the NLL term is assigned a smaller weight, this does not mean it plays a minor role, since its effect also depends on the loss scale, gradient magnitude, and learning objective. We set the NLL weight to 0.1 to achieve a better balance among distribution modeling, point forecasting accuracy, and training stability.
>
> **Q.3 Training cost analysis:**
>
> **A.5**: We thank the reviewer for this suggestion. Our method avoids the costly multi-step denoising process at inference time, but the multi-sample output used during training does introduce additional computational overhead. Section 6.5 provides an explicit theoretical analysis of the computational complexity of our method, and we further provide a training cost analysis under the same setting (Traffic, batch size =1) in the table(s). **We agree that this should be viewed as a trade-off: our method improves inference efficiency, while incurring extra training cost due to multi-sample prediction.**
>
> | Method   |  PPm  | NsDiff | TMDM  | TimeGrad |
> |:--------:|:-----:|:------:|:-----:|:--------:|
> | Training(s) | 1.16  |  1.84  | 0.88  |  4.23    |
> | Testing(s)  | 2.54  |  2.34  | 11.20 |  34.92   |
>
> **Q.4: The meaning of Gauss. and Unif.:**
>
> **A.6**: Thank you for your correction.“Gauss.” and “Unif.” indeed refer to using only parameterized Gaussian and uniform distributions, respectively, for probabilistic forecasting.

---

> > ### Author Rebuttal · Reviewer_PZr9 · 2026-04-01
> >
> > > **Question for A.1**
> >
> > As I understand it, Theorem 5.1 mainly establishes properties of the training objective, rather than directly demonstrating that the learned conditional distribution converges to the true $p(y \mid x)$. In particular, the connection between minimizing the proposed objective and recovering the true conditional distribution (where each conditioning input $x$ has only one observed target $y$) does not appear to be explicitly established. If this interpretation is inaccurate, I would appreciate further clarification.
> >
> > > **Question for A.2**
> >
> > While the distinction is clearer, it remains unclear what the theoretical or empirical motivation is for removing posterior modeling. Is there evidence that removing posterior inference leads to improved performance or better modeling compared to standard encoder-decoder latent variable models?
> >
> > > **Follow-up for A.5**
> >
> > Thank you for the clarification. This trade-off between improved inference efficiency and increased training cost is reasonable; however, it would have been more appropriate to explicitly acknowledge it in the limitations section.
> >
> > > **Question for A.6**
> >
> > Thank you for the clarification. However, I am still unclear about the setup for "Unif.". Does this mean that probabilistic forecasting is performed using a parameterized uniform distribution? If so, what are the parameters; are they the support bounds (lower and upper endpoints), or some alternative parameterization?

---

> > > ### Author Response · Authors · 2026-04-02
> > >
> > > **Q.1: Theorem and Connection**
> > >
> > > **A.1:**
> > >
> > > **Concern.1**: Yes, your interpretation is correct. Theorem 5.1 mainly establishes the consistency properties of the training objective. However, your understanding is partially right. **PPM is supported by Theorems 5.1 and 5.2 together to approximate the true conditional distribution.**
> > >
> > > **(1) Theorem 5.1 addresses training objective consistency.**
> > >
> > > It shows that, under finite K and h, the KDE-approximated NLL serves as a reliable proxy for the ideal conditional NLL, thereby justifying the objective we optimize. In this sense, Theorem 5.1 provides optimisability.
> > >
> > > **(2) Theorem 5.2 addresses model expressiveness/approximation ability.**
> > >
> > > It shows that the conditional distribution family induced by the push-forward map is sufficiently expressive to approximate the true conditional distribution $p^*(y|x)$. In this sense, Theorem 5.2 provides approximability.
> > >
> > > **Concern.2:**  For the concern regarding "connection between ...", we solve this concern (each conditioning input  has only one observed target) by the following designs:
> > >
> > > **(1) Constructing an input-aware conditional prior $p_{\theta}(z|x)$:**
> > >
> > > This procedure is not carried out independently for a single sample, but learned jointly over the whole dataset. For each given input x, the encoder first produces an input-aware parametric prior distribution $p_\theta(z | x)$, rather than a single output.
> > >
> > > **(2) Forming an empirical approximation of the predictive distribution $q_{\theta,\phi}(\cdot | x)$:**
> > >
> > > From this prior distribution, we then draw **K latent samples $(z^{(k)})^K_{k=1}$** (reparameterization), and map them through the push-forward function $g_\phi$ to obtain **K predictive samples $(\hat{y}^{(k)})^K_{k=1}$**. These predictive samples form a Monte Carlo empirical approximation of the conditional predictive distribution $q_{\theta,\phi}(\cdot \mid x)$.
> > >
> > > **(3) Approximating the conditional likelihood by KDE and training via Maximum Likelihood:**
> > >
> > > Given the observed target $y$ associated with input $x$, we use KDE over these K predictive samples to estimate the conditional density $q_{\theta,\phi}(y \mid x)$, and **maximize the corresponding log-likelihood.** Here, the one observed $y$ can be viewed as one realization from the unknown true conditional distribution $p^*(y \mid x)$, and therefore provides a valid supervision signal for conditional likelihood learning.
> > >
> > > Through these designs, we can construct this connection.
> > >
> > > **Q.2: Motivation and Evidence for removing posterior modeling**
> > >
> > > **A.2:**
> > >
> > > The motivation for removing explicit posterior modeling is to streamline the generative path, specifically mitigating the vulnerability of standard latent variables to **rapid distributional shifts (e.g., drifting means and dynamic variances)** in non-stationary environments. This eliminates the training-inference mismatch while ensuring theoretical convergence and computational efficiency.
> > >
> > > **1. Theoretical Support:** Theorems 5.1 and 5.2 together guarantee that the architecture can converge to the true conditional distribution $p^*(y|x)$ as illustrated in A.1.
> > >
> > > **2. Design Rationale:**
> > >
> > > **Streamlining the Generative Path:** Standard latent variable models (e.g., VAEs) inherently suffer from a training-inference mismatch—relying on a future-aware posterior $q(z|x,y)$ during training, but falling back to a prior $p(z|x)$ during inference. PPM streamlines this by maintaining a consistent, direct-mapping path in both stages, entirely avoiding this mismatch in highly non-stationary environments.
> > >
> > > **Structure-Aware Prior:** By constructing a structure-aware prior rather than generating from generic Gaussian noise, the model significantly reduces transport complexity. This eases the learning burden, captures complex temporal dynamics more effectively, and delivers a $2\times$ to $100\times$ inference speedup compared to leading iterative generative baselines with a single push-forward mapping.
> > >
> > > **3. Empirical Validation:**
> > >
> > > **Performance:** The framework achieves state-of-the-art results across seven real-world benchmarks.
> > >
> > > **Ablation Results:**  As shown in our previous rebuttal, replacing the variance of the learned prior variance with an identity covariance matrix leads to a significant decline in performance, justifying the necessity of the proposed design over standard latent refinement.
> > >
> > > **Q.3: Trade-off**
> > >
> > > **A.3:**
> > >
> > > Thank you for the suggestion. We will correct this in the revised version.
> > >
> > > **Q.4: Uniform**
> > >
> > > **A.4:**
> > >
> > > Yes, “Unif.” refers to using a parameterized uniform distribution. Since a uniform distribution can be parameterized in different ways, we clarify our implementation here. We predict its mean μ and log-variance, following a parameterization style consistent with the Gaussian case. We then derive the support interval as $[μ−3σ,μ+3σ]$, where $σ$ is obtained from the predicted log-variance, and sample the uniform prior from this interval.

---

### Official Review · Reviewer_s4HN · 2026-02-28

**Soundness:** 2
**Presentation:** 3
**Significance:** 3
**Originality:** 3
**Overall Recommendation:** 4
**Confidence:** 4

**Summary:**

The paper introduces Parametric Prior Mapping (PPM), a framework for non-stationary probabilistic multivariate time series (MTS) forecasting. The method encodes historical context into the sufficient statistics (mean and variance) of a Gaussian "prior." It then uses the reparameterization trick to sample latent vectors, which are pushed forward via a channel-independent MLP to generate future trajectories. The model is trained using a hybrid objective consisting of a Kernel Density Estimation (KDE)-based Negative Log-Likelihood (NLL) applied exclusively to 1D marginals, along with an auxiliary Mean Squared Error (MSE) on the sample mean. The authors claim this approach combines the efficiency of parametric priors with the expressiveness of deep generative models.

**Compliance With Llm Reviewing Policy:**

Affirmed.

**Final Justification:**

Thank for a thorough and well-organized rebuttal that addresses the majority of my concerns with substantive experiments.Although the theoretical framework has known limitations, these limitations have been honestly acknowledged and do not affect the practical value of the method. I am happy to raise my score given the substantial experimental additions. I would ask that the final manuscript explicitly acknowledge the marginal-only nature of the KDE-NLL objective and its implications for joint distributional quality, as this is an important caveat for practitioners using PPM in settings where temporal coherence of samples matters.

**Key Questions For Authors:**

1.The push-forward MLP gϕ operates per-channel (channel-independent). Does this mean the predictive distribution factorizes as $p(y_1,...,y_C|x) = \prod p(y_c|x)$? If so, cross-variable uncertainty correlations are not modeled. Can the authors confirm this, and provide results or discussion on multivariate calibration metrics (e.g., energy score) that capture joint distributional quality?

2.After z-score normalization, the scale of residuals varies substantially across datasets. Could the authors provide the empirical distribution of residuals ${y\_{c,t} - \hat y^{(k)}\_{c,t}}$ for different datasets, and confirm that $h=0.3$ is in the valid operating regime (i.e., neither severely over- nor undersmoothing) for all seven datasets?

3.Why are DeepAR and TFT not included in Tables 2 and 3? The paper's core narrative centers on combining parametric efficiency with generative expressiveness, yet no purely parametric probabilistic baseline is compared. Would the authors be willing to add these comparisons?

4.Theorem 5.2 assumes Assumption A.12 (dense MLP in $L^1(p_\theta(·|x))$). The practical implementation uses a two-layer MLP. Could the authors clarify what capacity this two-layer MLP actually has in terms of approximating the transport map, and whether there are cases (e.g., highly multimodal target distributions) where the two-layer architecture would be expected to fail?

5.Appendix B.1 states the sliding stride is set to the prediction length for large-scale datasets. How many test windows does this yield for Traffic and Electricity specifically? Are the baseline results in Tables 2 and 3 reproduced under this same protocol from scratch, or are they taken from original papers? If reproduced, are the numbers consistent with published results?

6.Why is only L=192 reported? Standard benchmark evaluation typically covers multiple horizons. Do PPM's gains diminish for shorter horizons (e.g., L=24, L=96)? This is particularly relevant for the non-stationarity claim, since short-horizon predictions are less affected by distributional shift.

7.What prevents σ from collapsing toward zero during training? Could the authors provide empirical histograms of σ values on the test sets, and confirm that the predictive intervals are not degenerate? Could they also compare PPM with a version that includes KL regularization to $N(0,I)$?

8.The KDE-NLL in Eq. 6 is a sum of marginal log-likelihoods. Could the authors report Energy Score or Variogram Score on at least two datasets, to assess whether PPM's generated samples capture temporal and cross-variable correlation structure? If the Energy Score is substantially worse than CRPS/QICE rankings suggest, this would indicate that the gains are marginal-only.

9.Regarding the MI lower bound analysis (Figure 5): The x-axis appears to be batch index over the test set. Could the authors clarify what is being averaged within each "batch index" point, and why variability across batches rather than a single aggregate MI estimate is presented? How is PPM's MI lower bound computed — does it use the same estimator as TMDM and NsDiff?

10.Regarding the cited reference Finzi et al., 2026: This is an arxiv preprint from January 2026. The paper uses it to motivate using structure-aware priors via "information-theoretic insights." Could the authors specify exactly which result from Finzi et al. is being invoked, and whether the paper's conclusions hold independently of this citation?

For other questions, please see weaknesses.If the authors can resolve these questions during the rebuttal, I am highly open to raising my score.

**Limitations:**

yes

**Strengths And Weaknesses:**

Strengths

1.The core design choice — constructing a context-aware parametric prior and transporting it via a lightweight push-forward map rather than running a multi-step denoising chain — is simple, well-motivated, and practically sensible. The identification that TMDM and NsDiff fail to capture time-varying aleatoric uncertainty due to rigid prior initialization (Figure 1) is a clear and well-illustrated motivating observation. The KDE-based NLL training objective for an implicit generative model, combined with the MSE mean-anchoring term, is a coherent design that avoids the need for an explicit density parameterization while remaining computationally tractable. The theoretical analysis, particularly the finite-(K,h) error decomposition (Theorem 5.1/A.7) and the gradient complementarity result (Theorem A.18), is one of the stronger theoretical contributions in this area of the time series literature.

2.The ablation study in Table 5 clearly separates the contributions of the NLL and MM terms and confirms that each serves a distinct role (distributional calibration vs. point accuracy). The prior robustness analysis (Tables 4, 8) is thorough, testing Gaussian, Uniform, Laplace, Student-t, Logistic, and Gumbel priors — a genuinely unusual and valuable experiment that demonstrates that the push-forward mapping's effectiveness is not contingent on a well-matched prior. The end-to-end training analysis (Table 9) confirms that joint optimization outperforms two-stage decoupled training, particularly for calibration. The sensitivity analysis for $K, \alpha, h,$ and backbone architecture is comprehensive. Inference time comparisons in Figure 6 are practical and informative.

3.The paper is well-organized. Figure 2 clearly illustrates the three-stage training pipeline. The theoretical section is formally structured with lemmas, theorems, and proofs in the appendix. The main text is concise and the writing is generally clean.
Significance of contributions: The inference speed advantage (2×–100× over diffusion models) is practically significant for real-world deployment. The framework is backbone-agnostic (demonstrated with MLP, NsTransformer, iTransformer in Table 7), which increases its practical utility. The non-stationarity framing using Fourier spectrum variance (Table 1) is a useful and underused diagnostic for characterizing dataset difficulty.

Weaknesses

Critical Flaws（1-8）

1.The transport map $g\_\phi$ is a two-layer channel-independent MLP, which does not model cross-variable or cross-temporal dependencies in the push-forward step. The latent code $z \in \mathbb{R}^{C×D}$ is processed per-channel, and $g\_\phi$ maps from D-dimensional latent to L-dimensional forecast independently per channel. This means the push-forward stage cannot capture any cross-channel uncertainty structure in the predictive distribution, e.g., correlated errors between variables. For univariate-per-channel generation this may be sufficient, but PPM is presented as a framework for multivariate time series, and the full multivariate predictive distribution $ p(y_1, ..., y_C | x)$ is not modeled jointly. The paper is largely silent on this limitation.

2.The KDE bandwidth h is a fixed global hyperparameter ($h=0.3$ for all datasets), which is theoretically suboptimal. In Eq. 13, the bound shows an explicit bias-variance tradeoff: smaller h reduces smoothing bias but amplifies finite-K fluctuation. A single fixed h across heterogeneous datasets (ETTh1 variance 3.69, Weather variance 0.387 — nearly 10x range) means the KDE is implicitly operating in very different regimes. The paper acknowledges bandwidth sensitivity in Appendix C.5.3 and shows that performance degrades outside an optimal range, but does not explain why a single $h=0.3$ is simultaneously optimal across all datasets. In practice, the effective role of KDE with a fixed bandwidth on different-scale datasets after z-score normalization is unclear.

3.Theorem 5.2 (Universality) relies on Assumption A.12 (Universal approximation in $L^1$), which assumes the MLP class is dense in $L^1(p_\theta(·|x))$ for any measurable target map $T\_x$. This is a standard universality result for MLPs, but it assumes unlimited network capacity. In the practical implementation, $g\_\phi$ is a two-layer MLP — this is architecturally quite constrained. The theorem therefore provides a theoretical existence guarantee that may not be informative about the finite-capacity behavior of the actual model. The paper does not discuss the gap between the theoretical universality claim and the two-layer MLP actually used.

4.Missing regularization on the learned variance and risk of uncertainty collapse. The encoder outputs $\sigma$ via a Softplus activation (ensuring $\sigma > 0$), but there is no explicit regularization term — such as a KL divergence to a standard prior as in the VAE formulation — to prevent σ from collapsing toward zero during training. In principle, the KDE-NLL can be maximized at a given target y by driving all K samples {$\hat y\^{(k)}$} to concentrate near y, which in turn is facilitated by $\sigma \rightarrow  0$. The MSE mean-anchoring term (MM) does not prevent this, as it is equally well minimized by a degenerate distribution. While the fixed KDE bandwidth $h$ provides some implicit resistance to full Dirac collapse by smoothing the density estimate, this is a fragile, bandwidth-dependent safeguard rather than a principled regularization mechanism. The paper provides no theoretical bound on σ and no empirical analysis of the distribution of σ values on the test sets, leaving the uncertainty preservation claim unsupported.

5.The KDE-NLL objective supervises only univariate marginals, not the joint distribution. Equation 6 decomposes as a sum of per-coordinate log-densities $\sum\_c \sum\_t \log q_{\theta,\phi}(y_{c,t}|x)$, where each term involves a 1D KDE estimate over K scalar samples. This objective provides no gradient signal for temporal autocorrelation structure or cross-channel dependencies — a model generating temporally i.i.d. noise that matches the correct 1D marginals would achieve identical $L\_{NLL}$ to a model with correct temporal structure. Combined with the channel-independent push-forward mapping (Weakness 1), PPM is structurally equivalent to a collection of C×L independent univariate density estimators. This represents a fundamental limitation for probabilistic multivariate forecasting, where the joint predictive distribution is what matters for downstream decision-making.

6.Evaluation metrics are exclusively univariate marginal statistics, masking the joint distribution failure. Both CRPS and QICE are computed per scalar coordinate and averaged, making them incapable of detecting failures in temporal or cross-channel joint distribution modeling. A model that generates temporally uncorrelated noise with correct marginals would score identically to a true joint generative model on these metrics. Multivariate proper scoring rules — specifically the Energy Score and Variogram Score — are necessary to validate joint distributional quality. The absence of these metrics, combined with the marginal-only training objective, means the evaluation setup is structurally aligned with PPM's limitations and cannot serve as evidence for the joint probabilistic forecasting claims. This gives PPM an unfair advantage over baselines like DiffusionTS and TMDM that do model joint distributions.

7.The reparameterization trick (Eq. 4) samples $z = \mu + \sigma \odot \varepsilon$ with z ∈ R^{C×D} and $\varepsilon \sim N(0,I)$. The latent dimension D is set to 128 (Table 6) and L=192. The mapping $g\_\phi$: $\mathbb {R}^{C×D} → \mathbb {R}^{L×C}$ effectively learns a $D \rightarrow L$ projection per channel. Since D=128 < L=192, this is a dimension-expanding map. It is unclear whether this dimensionality choice is principled: why is D=128 sufficient to encode sufficient statistics for generating 192-step forecasts? The paper does not analyze the latent dimension sensitivity beyond Table 6 footnote.

8.The model only evaluates a single fixed prediction horizon of L=192 with H=96. Standard benchmarks for probabilistic forecasting typically evaluate multiple horizons (e.g., {24, 48, 96, 192} for ETT datasets). Fixing L=192 hides whether PPM's advantage holds across short-horizon settings where non-stationarity is less pronounced. The prior-based initialization may be less beneficial for short horizons, and demonstrating performance across a range of L would strengthen the claims considerably.

9.All baselines are diffusion-based generative models, and no parametric probabilistic forecasting baselines are included. The paper positions PPM as combining parametric efficiency with generative expressiveness, but DeepAR, BetterDeepAR, and TFT (all cited in the related work) are absent from the comparison tables. This makes it impossible to assess whether PPM actually improves over strong parametric baselines, or whether the gains are relative to a set of comparably weak diffusion baselines. Given that the paper's core argument is about the synergy between parametric and generative approaches, this omission is significant.

10.The comparison with NsDiff — the most directly relevant baseline — is complicated by a potential fairness issue. NsDiff uses sliding-window variance computed over the training data as a prior, while PPM learns the prior dynamically. However, it is unclear whether PPM's encoder fθ and NsDiff's variance module are trained with the same number of parameters and similar computational budgets. The inference-time speedup comparison in Figure 6 shows NsDiff as markedly slower, but the paper does not provide model parameter counts for all baselines, making this comparison incomplete.

11.The reported improvements are stated as "up to 31.2% reduction in CRPS" and "44.3% in QICE" but these appear to be best-case figures across datasets. For example, on ETTh2, NsDiff achieves CRPS 0.349 and PPM achieves 0.306 — a 12.3% improvement, substantially below 31.2%. The 31.2% figure appears to correspond to the Traffic dataset specifically. Presenting "up to X%" figures prominently in the abstract and contributions without specifying which dataset achieves the maximum improvement is potentially misleading.

12.The ablation in Table 5 only covers two datasets (ETTm1, Electricity) and examines only the objective function decomposition. There is no ablation on the encoder architecture fθ, the latent dimension D, or the number of prior samples K used in training (though sensitivity analyses are provided separately). The component-level ablation is thus incomplete relative to the full model design.

13.The sliding stride in evaluation for large-scale datasets (Electricity, Traffic, Weather) reduces the number of evaluated windows. Appendix B.1 states that for these datasets, "we set the sliding stride of the sampling window to the prediction length." This substantially reduces the number of test samples compared to evaluations in the literature that use stride-1 sliding windows. The authors state all baselines use this same setting, but this protocol differs from most prior work, making direct comparison with published numbers from other papers unreliable. The paper should clarify this explicitly.

14.The main text states "The parameters of KDE-NLL loss, including bandwidth h and loss weighting α, are set to 0.3 and 0.1, respectively, for all datasets." However, Appendix C.5.3 shows that the optimal h varies between datasets (the optimal appears to be around 0.2-0.4 depending on the dataset). The claim that h=0.3 is universally good is somewhat undermined by the sensitivity analysis itself.

15.Figure 5 shows mutual information lower bounds across "batch indices" for the test set. The x-axis label "batch index" suggests this is plotted per mini-batch of the test set, not averaged. This is an unusual visualization choice — the variability across batches (not across epochs) is being shown, which conflates the MI estimate's variance due to finite batch size with actual differences in information content. The comparison with baselines (TMDM, NsDiff) via MI lower bound is not the standard way to compare prior conditioning quality and the validity of this comparison is not established.

16.The paper does not compare with flow matching approaches for time series (e.g., TSFlow, Kollovieh et al., 2024, which is cited in the related work but not used as a baseline). The push-forward formulation of PPM is architecturally closer to flow matching than to diffusion, yet no flow matching baseline is included.

17.K2VAE (Wu et al., 2025), which also addresses non-stationarity in probabilistic forecasting via Koopman theory, is not mentioned. Given that it was published at ICML 2025 and directly relevant to the problem setting, its omission is notable.

18.The paper does not discuss calibration beyond CRPS and QICE. Reliability diagrams or empirical coverage probabilities at multiple quantile levels would provide a more complete picture of distributional calibration, especially for the non-stationarity claim.

---

> ### Author Rebuttal · Authors · 2026-03-31
>
> We sincerely thank the reviewer for the detailed and professional comments. Due to the rebuttal length limit, we focus here on the major concerns. In addition, some tables and figures for answer are provided in the anonymous link https://justpaste.it/gjswv.
>
> **W.1, W.5, W.6, Q.1, Q.8 Joint distribution modeling:**
>
> **A.1:**
>
> **Our model is a framework for multivarite time series.**  Although $f_{\theta}$ and $g_{\phi}$ do not explicitly model cross-variable relationships, channel-independent modeling is still a common strategy in multivariate ts forecasting. Meanwhile, in Section C.1 of the paper, we analyze the effect of replacing different$f_{\theta}$. It can be seen that when$f_{\theta}$is replaced by iTransformer, the performance improves significantly. This indicates that cross-variable relationships can still be learned through the conditional encoding network.
>
>  **Independent-point learning is an effective strategy for time series forecasting.** We agree that the KDE loss is a collection of $C\times L$ estimators and does not explicitly model the joint distribution in termporal and variables. **However, for a highly conditioned time-series forecasting task, each prediction point can make use of the temporal and cross-variable relationships learned from the historical window.** Similarly, our current loss can also be viewed in the same spirit as the widely used MSE loss in point forecasting: although MSE is also essentially a collection of point-wise losses, models such as DLinear, iTransformer, and PatchTST still achieve strong performance with MSE loss.
>
> We also present the Energy Score results in the table below, which show that our method can still learn conditional joint distribution information to some extent. For example, **on the Traffic dataset, where joint dependence is stronger, PPM also achieves substantial improvements in terms of Energy Score.**
>
> Additionally, we acknowledge that joint distribution modeling is worth consideration, as also pointed out by FreDF. To further investigate this, we conduct an experiment by replacing the KDE loss with Energy Score loss within the PPM framework (PPM-Escore in Table). **The results show further improvements on ETTh2 and Weather with energy score,** although on Traffic, it leads to out-of-memory. We will further study this issue in future work.
>
> | Dataset    |    ETTh2     | ETTh2 |   Weather    | Weather |   Traffic    | Traffic |
> |:--:|:---:|:--:|:--:|:--:|:--:|:--:|
> |   Metric   | E-Score | CRPS  | E-Score |  CRPS   | E-Score |  CRPS   |
> |   NsDiff   |    16.975    | 0.350 |    22.397    |  0.253  |   237.147    |  0.367  |
> |    PPM     |    15.744    | 0.306 |    22.177    |  0.215  |   **208.448**    |  **0.252**  |
> | PPM-Escore |    **15.166**    | **0.300** |    **21.358**    |  **0.210**  |     OOM      |   OOM   |
>
> **W.4 and Q.7 Regulazition for $\sigma$:**
>
> We agree that potential collapse of $\sigma$ is important. The histograms in the link Fig.1 of ETTh1 and Electricity show no obvious collapse. This is expected because our objective is to generate a sample set that covers future distribution; **if $\sigma$ collapsed, the samples would become concentrated and be a Dirac distribution, leading to an increase in KDE loss.**
>
> We also evaluated a KL regularization term toward N(0,1), as reported in the table. It slightly improves MSE on ETTh2, but clearly worse on QICE; on Electricity, it improves QICE but does not improve point forecasting performance. **Overall, explicit variance regularization does not yield consistent improvements, and its effect is dataset-dependent.**
>
> | Dataset  |  ETTh2 | ETTh2 | ETTh2 | Elec| Elec| Elec|
> |:--:|:--:|:--:|:--:|:--:|:--:|:--:|
> | Metric   |    MSE  | CRPS  | QICE  |     MSE     |    CRPS     |    QICE     |
> | +KL_loss | 0.370 | 0.319 | 4.155 |    0.185    |    0.205    |    1.678    |
> | PPM      |  0.376 | 0.306 | 2.011 |    0.182    |    0.206    |    2.435    |
>
> **W.2, W.14, Q.2  Parameters h:**
>
> Yes, you are right. We agree that describing h=0.3 as universally optimal is not sufficiently precise; a shared bandwidth was mainly intended to demonstrate cross-dataset stability and avoid hyperparameter tuning. Following your suggestion, we examine the residual histograms in the link Fig.2 and found that except for Weather, h=0.3 remains within a relatively reasonable operating regime for most datasets.
>
> **W.13, Q.5 Sliding stride in eval:**
>
> We follow the evaluation protocol of stride length in NsDiff.
>
> **W.15, Q.9 MI lower-bound:**
>
> Yes. We use the same MI lower-bound estimator. Figure 5 is not meant to show training fluctuation, but to examine conditional information capture along the test-time axis. Since test batches are ordered sequentially without shuffling, different batch indices correspond to different temporal stages. **Under this setup, the figure reflects how the model’s conditional information capture changes over time, which is relevant under non-stationary distributions.**

---

> > ### Author Rebuttal · Reviewer_s4HN · 2026-04-02
> >
> > Thank you for a thorough and well-organized rebuttal that addresses the majority of my concerns with substantive experiments.Although the theoretical framework has known limitations, these limitations have been honestly acknowledged and do not affect the practical value of the method. I am happy to raise my score given the substantial experimental additions. I would ask that the final manuscript explicitly acknowledge the marginal-only nature of the KDE-NLL objective and its implications for joint distributional quality, as this is an important caveat for practitioners using PPM in settings where temporal coherence of samples matters.

---

> > > ### Author Response · Authors · 2026-04-02
> > >
> > > Thank you for the review and all the suggestions! We will clarify in the final manuscript the limitations of PPM in modeling the joint distribution.

---

### Official Review · Reviewer_pDvE · 2026-03-09

**Soundness:** 3
**Presentation:** 3
**Significance:** 3
**Originality:** 2
**Overall Recommendation:** 4
**Confidence:** 3

**Summary:**

The study proposes the Parametric Prior Mapping (PPM) framework for probabilistic multivariate time-series forecasting. The model can be interpreted as a conditional latent-variable generator. An encoder network takes the historical sequence $x$ and produces the parameters $\mu(x)$ and $\sigma(x)$ of a Gaussian latent distribution $z \sim \mathcal{N}(\mu(x), \sigma(x))$. Multiple latent samples $z_1,\ldots,z_K$ are drawn from this conditional distribution and passed through a decoder network that has the task of generating future trajectories $y_1,\ldots,y_K$ (conditioned on $z$ and, therefore, on the input $x$). These generated samples form an empirical approximation of the predictive distribution $q(y \mid x)$. Training is performed using a cost function composed of two terms. The first is the negative log-likelihood (NLL) based on kernel density estimation (KDE), which encourages the observed future trajectory $y$ to lie in a high-density region of the generated samples. The second is an auxiliary MSE term that enforces the mean of the generated trajectories to be close to the observed trajectory. The paper also presents theoretical results analyzing the properties of the PPM model, and empirical evaluations on several benchmark datasets.

**Compliance With Llm Reviewing Policy:**

Affirmed.

**Final Justification:**

After reading the rebuttal, I remain at Weak Accept with increased confidence. The authors directly addressed my main empirical concerns by adding both broader-domain evaluations (Exchange-Rate and ILI) and comparisons with classical parametric probabilistic forecasting baselines (DeepAR, BetterDeepAR). The additional PPM-Gauss ablation is especially useful, as it cleanly isolates the benefit of the push-forward refinement beyond the parametric prior alone. My view on originality remains unchanged—the work is still more of an elegant synthesis than a fundamentally new paradigm—but the stronger empirical support makes the overall contribution more convincing.

**Key Questions For Authors:**

1 - Can the authors say something about how PPM behaves when the conditioning variable $x$ lies outside the training distribution (e.g., regime shifts or structural changes)?
2 - Could the authors comment on how PPM compares with classical statistical probabilistic forecasting methods?

**Limitations:**

Yes

**Strengths And Weaknesses:**

The authors propose PPM for time-series forecasting. Instead of starting generative models from generic noise such as $N(0,I)$, they start from a structured parametric distribution $N(\mu(x), \sigma(x))$  derived from the data. This injects strong inductive bias from classical forecasting models.  One advantage of PPM is that it generates samples in one pass. Another is that PPM  architecture is intentionally very simple, using only MLP rather than complicated transformer stacks. In summary, this is an incremental piece of research that is interesting and conceptually clear.

The theory section (and appendix) presents results describing the properties of the PPM model. These results do not require technically deep proofs and are not individually highly novel. However, their combination within this specific modeling framework is sufficiently original to be of interest. The proofs are carefully written, and the authors deserve credit for the clarity and rigor of their theoretical development. Overall, the theoretical results serve primarily to provide supporting justification and insight into the behavior of the proposed method rather than to introduce fundamentally new mathematical techniques.

The theoretical analysis focuses on three main aspects. The first analyzes the difference between the true NLL and its KDE-based approximation computed from samples. Theorem 5.1 shows that the approximation error decomposes into two components corresponding to the smoothing bias (controlled by the kernel bandwidth) and the finite-sample variance (controlled by the number of samples). This result applies standard kernel density estimation theory to the proposed objective.

Next, Theorem 5.2 establishes that the generated conditional distribution can approximate any target conditional distribution under mild assumptions. This follows from the universal approximation properties of neural networks together with standard results on push-forward distributions from optimal transport theory. In essence, the theorem shows that with a sufficiently expressive decoder $g_{\phi}$, the model class is dense in the space of conditional distributions.

Finally, the authors analyze the gradient of the KDE-based likelihood with respect to the generated samples. They show that gradients tend to become winner-take-all when the kernel bandwidth is small, and that the auxiliary MSE term provides a stable global gradient signal during optimization.

Although these results are not mathematically difficult to prove, they play an important role in justifying the KDE likelihood formulation. They establish the expressiveness of the model, and explain the optimization behavior of the training objective. Overall, the theoretical analysis clarifies the behavior of the proposed loss function and provides useful support for the modeling choices. The fact that the arguments are clearly written and carefully derived adds to their value.

One interesting aspect is the generalization capacity of PPM if the input $x$ lies far outside the training distribution. The paper does not explicitly address the distribution shift in the conditioning variable $x$. While the conditional prior $p(z|x)$ adapts to the input through a learned encoder, the method assumes that the test inputs lie within the support of the training distribution. In strongly nonstationary settings, where new regimes may appear at inference time, the quality of the predictive distribution may degrade. Of course, this is a fundamental limitation of all conditional generative models, and therefore should not be interpreted as a flaw specific to this paper.

The experimental section is a strong part of the paper. The authors compare their model against several strong generative forecasting baselines using appropriate evaluation metrics. PPM generally improves performance across datasets and also shows a clear speed advantage compared with diffusion-based models. One aspect that I missed is the presence of classical statistical baselines, as the comparisons are limited to deep generative models. I also missed a broader diversity of datasets suh as, for example, from finance or from epidemiology.

---

> ### Author Rebuttal · Authors · 2026-03-31
>
> **Q1. The work is incremental:**
>
> **A1.** We respectfully disagree that our work is merely incremental. As written in the paper contribution, PPM has three main innovations:
>
> **Data-driven prior distribution modeling**: which directly learns an adaptive prior from the input condition.
>
> **Resampling and push-forward mapping**: which directly generate samples in support of the predictive distribution.
>
> **An NLL+MSE hybrid objective**: which balances distribution modeling, point forecasting accuracy, and training stability.
>
> We believe these advancements make our contribution not only conceptually distinct but also practically significant.
>
> **Q2. Limited diversity of datasets**
>
> **A2.** Thank you for pointing this out. We additionally included the financial **Exchange-Rate** dataset and the epidemiological **ILI** dataset for comparison. The results show that PPM still achieves strong performance in these datasets.
> | Dataset | Exc| Exc |Exc |ILI  | ILI | ILI |
> |:---:|:---:|:---:|:---:|:---:|:---:|:---:|
> | Metric | MSE | MAE | CRPS | MSE | MAE | CRPS |
> | NsDiff | 0.338 | 0.380 | 0.304 | 3.075 | 1.074 | 0.931 |
> | PPM | 0.189 | 0.305 | 0.253 | 2.489 | 0.966 | 0.739 |
>
> **Q3.  x outside the training distribution:**
>
> **A3.** Thank you for this insightful question. We agree that this is an important and challenging scenario in probabilistic time series forecasting. In practice, most existing methods experience performance degradation when the conditioning variable x falls outside the training distribution, such as under regime shifts or structural changes. However, compared with approaches based on fixed priors or static uncertainty modeling, **PPM employs an input-dependent conditional prior $p_{\theta}(z|x)$, which gives it stronger adaptability under moderate distribution shifts**.
>
> **Q4. Compares with classical statistical probabilistic forecasting methods:**
>
> **A4.** We additionally include comparisons with DeepAR and BetterDeepAR, both of which are classical parametric forecasting models that assume the predictive distribution follows a specific prior distribution. In Section 6.4.1, we also conduct an ablation by directly using PPM’s parametric classical distribution as the predictive distribution, and the corresponding results are presented in the table below.
>
> | Dataset | ETTh1 | ETTh1 | ETTh1 | ETTm2 | ETTm2 | ETTm2 | Elec | Elec | Elec | Weather | Weather | Weather |
> |:---|:---:|:---:|:---:|:---:|:---:|:---:|:---:|:---:|:---:|:---:|:---:|:---:|
> | Metric | MSE | MAE | CRPS | MSE | MAE | CRPS | MSE | MAE | CRPS | MSE | MAE | CRPS |
> | deepar | 0.797 | 0.624 | 0.466 | 0.535 | 0.508 | 0.378 | 0.304 | 0.377 | 0.276 | 0.259 | 0.333 | 0.293 |
> | Betterdeepar | 0.897 | 0.686 | 0.500 | 0.564 | 0.546 | 0.406 | 0.365 | 0.428 | 0.313 | 0.347 | 0.399 | 0.321 |
> | PPM-guass | 0.481 | 0.459 | 0.351 | 0.250 | 0.309 | 0.237 | 0.190 | 0.272 | 0.211 | 0.248 | 0.273 | 0.218 |
> | PPM | 0.447 | 0.432 | 0.337 | 0.247 | 0.306 | 0.237 | 0.182 | 0.267 | 0.206 | 0.242 | 0.268 | 0.215 |
>
> The weaker performance of DeepAR and its variants may be partly attributed to the limited ability of recurrent neural networks to capture long-range dependencies. In addition, our ablation results further highlight the advantage of our method over classical statistical approaches.

---

> > ### Author Rebuttal · Reviewer_pDvE · 2026-04-03
> >
> > After reading the rebuttal, I remain at Weak Accept with increased confidence. The authors directly addressed my main empirical concerns by adding both broader-domain evaluations (Exchange-Rate and ILI) and comparisons with classical parametric probabilistic forecasting baselines (DeepAR, BetterDeepAR). The additional PPM-Gauss ablation is especially useful, as it cleanly isolates the benefit of the push-forward refinement beyond the parametric prior alone. My view on originality remains unchanged—the work is still more of an elegant synthesis than a fundamentally new paradigm—but the stronger empirical support makes the overall contribution more convincing.

---

> > > ### Author Response · Authors · 2026-04-04
> > >
> > > Thank you for your continued support and for increasing your confidence score. We are glad that the new baselines and ablations successfully addressed your empirical concerns.
> > >
> > > Regarding originality, we agree that proposing a fundamentally new paradigm is highly challenging, which we will continue striving toward in our future research.
> > >
> > > However, the concrete innovations of our work remain distinct and significant. While our method also pushes forward a prior to generate a predictive distribution, it introduces key advancements:
> > >
> > > 1. Our proposed multi-sample output, combined with the KDE loss, enables the direct maximization of log-likelihood for stable optimization.
> > >
> > > 2. The parametric prior, which is directly learned from data, adaptively captures complex non-stationary features, overcoming the limitations of heuristic priors in existing generative models.
> > >
> > > As validated by the strong empirical results, these designs successfully combine the efficiency of parametric methods with the expressiveness of generative models. We believe these improvements offer a valuable contribution to probabilistic time series forecasting.

---

### Official Review · Reviewer_D77y · 2026-03-10

**Soundness:** 3
**Presentation:** 3
**Significance:** 2
**Originality:** 2
**Overall Recommendation:** 4
**Confidence:** 3

**Summary:**

The manuscript proposes to incorporate historical information into forecasting by encoding historical data into some parametric distribution (e.g., normal distribution). During the training process, the authors propose to use the kernel-estimated density as loss function.  Numerical results demonstrate a good empirical performance.

**Compliance With Llm Reviewing Policy:**

Affirmed.

**Final Justification:**

I think the author settle my concerns, so I would like to raise my score.

**Key Questions For Authors:**

1. I hope the authors further explain the difference between this work and the diffusion-based learning, such as ``Flow matching with gaussian process priors for probabilistic time series forecasting.'' in the reference. One difference is that the common diffusion model uses cross entropy in training. However, the framework in Figure 2 seems similar to the diffusion model.


2. One issue of kernel density estimator is the curse-of-dimensionality, i.e., when the dimension is high, the convergence rate is the estimator is low. In the setup of the manuscript, this involves a large number of resampling during training. I wonder if the authors can make further discussions on this issue beyond section C.5.1, i.e., why a less precise estimation in density leads to a relative good outcome.  Besides, in Lemma A.5 and A.7, since the resample size is small (few hundreds), the convergence rate should incorporate dimension.




3. Lemma A.10, lack of reference about the claim.

**Limitations:**

Yes

**Strengths And Weaknesses:**

Strengths: I think the idea of leveraging kernel density estimator in optimization is an interesting practice, and the overall numerical performance is good.


Weakness: Please see the ``Questions'' part.

---

> ### Author Rebuttal · Authors · 2026-03-31
>
> **Q1. Difference between this work and diffusion-based learning**:
>
> **A1.** We agree that, from a high-level perspective, both PPM and diffusion-/flow-matching-based methods can be viewed as generating the target predictive distribution from a prior, so Figure 2 may appear similar in form. However, their core mechanisms are different. Diffusion models typically rely on an explicitly defined forward noising process and a learned reverse denoising process, while Flow Matching methods usually construct an intermediate probability path between the prior and target distributions and learn the corresponding flow field. In contrast, PPM does not explicitly construct such an intermediate path or use an iterative generation; instead, **PPM starts from a structured conditional prior induced by historical information and directly generates K samples for forecasting distribution with reparameterization and push forward**. For time-series data with complex dynamics and non-stationarity, this design alleviates the dependence on a predefined intermediate path prior and also enables faster inference
>
> More discussion comparing our method with latent-variable generative models can be found in our response A.2 to Reviewer PZr9.
>
> **Q2. Curse of dimensionality of KDE**:
>
> **A2.** I am very glad to clarify this point for the reviewers. The curse of dimensionality is indeed a well-known challenge for kernel density estimation. However, in our method, KDE is applied in a one-dimensional manner, which **does not suffer from the curse of dimensionality.** As described in Eq. (6), we calculate the NLL loss through
>
> $$-\frac{1}{C L}\sum_{c=1}^C \sum_{t=1}^L\log q_{\theta,\phi}({y}_{c,t}|x)$$
>
>  in every point for KDE loss,
> instead of
>
> $$-\frac{1}{C}\sum_{c=1}^C \log q_{\theta,\phi}({y}_{c}|x)$$
>
> for high-dimension sequence.
>
> Although this treatment does not explicitly model the joint distribution structure, in highly conditioned time-series forecasting, modeling point-wise conditional marginals can still capture an important part of the predictive uncertainty in practice. More analysis about joint distribution modeling can be found in our response A.1 to Reviewer s4HN.
>
> **Q3. Lemma A.10 lacks a reference about the claim.**:
>
> **A3.** Thank you for pointing this out. The claim in Lemma A.10 relies on a standard measurable transport / probability isomorphism argument, but we agree that the current presentation does not provide an appropriate reference. We will add a citation and clarify the construction of the measure-preserving map used in the proof.

---

> > ### Author Rebuttal · Reviewer_D77y · 2026-04-02
> >
> > I do not have further comments here.

---

> > > ### Author Response · Authors · 2026-04-02
> > >
> > > Thank you for your recognition of both our work and our responses. If you have any further questions, we would be happy to address them.

---

### Decision · Program_Chairs · 2026-04-30

**Decision:**

Accept (regular)

**Comment:**

The reviewers uniformly found this paper well written, the theory well presented, and the experiments carefully done.  Reviewer pDvE found the paper "more an elegant synthesis than a fundamentally new paradigm" (a sentiment echoed in some of the other reviews), but still found this a convincing contribution.  Reviewers s4HN and PZr9 had many questions in the initial round, but were mostly happy after the rebuttal.  The main remaining point noted by both these reviewers was the limitations of the KDE-approximated NLL.  The authors have acknowledged that this is a proxy measure meant to balance optimization quality and computational performance, but that (for example) it only involves marginals.

After an extensive back and forth, three of the reviewers favored acceptance and one favored rejection (all weakly).  During the post-rebuttal discussion among reviewers, there was additional support for acceptance.  Given the generally positive feeling of the reviewers (and my own pleasure in reading), I am in support of accepting the paper as well.